
# kCARTA : A fast pseudo line-by-line radiative transfer algorithm with analytic Jacobians, fluxes, Non-Local Thermodynamic Equilibrium and scattering for the infrared

Sergio DeSouza-Machado[1], L. Larrabee Strow[1,2], Howard Motteler[1], and Scott Hannon[3]

[1]JCET, University of Maryland Baltimore County, Baltimore, Maryland
[2]Dept of Physics, University of Maryland Baltimore County, Baltimore, Maryland
[3]Deceased

**Correspondence:** Sergio DeSouza-Machado (sergio@umbc.edu)

**Abstract.** A fast pseudo-monochromatic radiative transfer package using a Singular Value Decomposition (SVD) compressed atmospheric optical depth database has been developed, primarily for use with hyperspectral sounding instruments. The package has been tested extensively for clear sky radiative transfer cases, using field campaign data and satellite instrument data. The current database uses HITRAN 2016 line parameters and is primed for use in the spectral region spanning 605 cm$^{-1}$ to

2830 cm$^{-1}$ (with a point spacing of 0.0025 cm$^{-1}$), but can easily be extended to other regions. The clear sky radiative transfer model computes the background thermal radiation quickly and accurately using a layer-varying diffusivity angle at each spectral point; it takes less than 20 seconds (on a 2.8 GHz core using 4 threads) to complete a radiance calculation spanning the infrared. The code can also compute Non Local Thermodynamic Equilibrium effects for the 4 $\mu m$ CO$_2$ region, as well as analytic temperature, gas and surface jacobians. The package also includes flux and heating rate calculations, and an interface

to a scattering model.

## 1   Introduction

Recent years have seen the launch and routine operation of new generation infrared sounders on board Earth orbiting satellites, for the purposes of providing measurements for data assimilation into Numerical Weather Prediction (NWP) centers and for monitoring atmospheric composition. These hyperspectral instruments have low noise channels with high resolution,

and provide gigabytes of data daily, from about 620 - 2800 cm$^{-1}$. Examples include the Atmospheric InfraRed Sounder (AIRS)(Aumann et al., 2003) on board NASA's Aqua satellite, the Infrared Atmospheric Sounding Interferometer (IASI) on board the Metop satellites (Clerbaux et al., 2009) and the Cross Track Infrared Sounder (CrIS) on board the Suomi and JPSS-1 satellites (Han et al., 2013).

The radiances measured by these instruments are obtained under all-sky conditions (*i.e.* clear or cloudy). Publicly available

thermodynamic profiles retrieved from this voluminous data is presently performed *after* cloud-clearing the radiances (Susskind et al., 1998). Monochromatic line-by-line (MNLBL) codes are too slow for use in the operational retrievals from the cloud-cleared radiances. Instead, optical depths (or transmittances) produced by these MNLBL codes are parametrized for use in



fast Radiative Transfer Algorithms (RTAs), at the instrument resolution. The accuracy of the retrieved products depends on the accuracy of the fast models, which underlines the importance of the accuracy of line parameters and lineshapes used in
MNLBL codes, particularly the far-wing effects.

Using MNLBL codes to produce optical depths for training the fast models is computationally intensive, as spectral resolutions of 0.0025 cm$^{-1}$ or better are required over the almost 2500 cm$^{-1}$ span of a typical infrared sounder, for 50 or more profiles. The acceleration of this part of the process needed to develop a fast RTA for the AIRS sounder, was the motivating factor behind the development of the work presented here. For this we also developed a line-by-line code (referred to as
UMBC-LBL) to produce an accurate pre-computed database of monochromatic atmospheric optical depths. Singular Value Decomposition (SVD) was then used to produce a highly compressed database (referred to as the kCompressed database (Strow et al., 1998)) that is highly accurate, relatively small, and easy to use. When coupled to an accurate radiative transfer code, it can be used as a starting point for developing tools for atmospheric retrievals (Rodgers, 2000).

To compute optical depths and radiances at any level for an arbitrary Earth atmospheric thermodynamic + gas profile, we
paired together an uncompression algorithm for the kCompressed database with a one dimensional clear sky radiative transfer algorithm (RTA). The RTA works for both a downlooking and an up-looking instrument, with geometric ray tracing accounting for the spherical atmospheric layers. The generation of monochromatic transmittances from the compressed database is orders of magnitude faster than using a MNLBL code; for the long paths in the atmosphere the computed transmittances are smooth and well behaved, and can be used to develop fast forward models. Radiances computed using the compressed database are
as accurate as those computed with a MNLBL code as our compression procedure introduces errors well below spectroscopy errors (Strow et al., 1998).

The entire package is called kCARTA, which stands for "kCompressed Atmospheric Radiative Transfer Algorithm". Although kCARTA *is* much slower than fast forward models which use effective convolved transmittances, it is much more accurate, and can be used to generate optical depths and transmittances for developing the faster models. An example is the
Stand Alone Radiative Transfer Algorithm (SARTA) (Strow et al., 2003) for which kCARTA is the Reference Forward Model; SARTA is used to retrieve Level 2 geophysical products from the AIRS (Strow et al., 2003) and CriS (Gambacorta, 2013) instruments. Other fast forward models for the infrared which parametrize the transmittances of the finite width instrument channels include a Principal Component based Radiative Transfer Model (PCRTM; (Liu et al., 2006)), Radiative Transfer for TIROS Operational Vertical Sounder (RTTOV; (Saunders et al., 1999) and the Jülich Rapid Spectral Simulation Code (JURASSIC;
(Hoffman and Alexander, 2009)).

kCARTA also includes algorithms to rapidly compute analytic jacobians, and is available in a Fortran 90 package. This package (v1.21, April 2019) uses some of the newer Fortran features such as implicit loops, function overloading and modules, and includes code for computing fluxes, heating rates, and the effects of cloud and aerosol scattering using the Parametrization of Clouds for Longwave Scattering in Atmospheric Models (PCLSAM) (Chou et al., 1999) algorithm. While kCARTA was
developed for use in the infrared region (605-2830 cm$^{-1}$), it is trivial to extend the database out in either direction, to span the Far InfraRed to the Ultra-Violet. A clear-sky only radiance+jacobian Matlab version is also available.



The speed and accuracy, plus available run-time options of the code make it a very attractive alternative to other existing line by line codes. The literature is replete with papers and books describing spectroscopic calculations, monochromatic radiative transfer and flux calculations (see for example (Goody and Yung, 1989; Edwards, 1992; Clough et al., 1992; Clough and Iacono,

1995; Tjemkes et al., 2002; Buehler et al., 2011; Schreier et al., 2014; Dudhia, 2017; Vincent and Dudhia, 2017) so here we chose to emphasize the features (and limitations) of kCARTA that would interest researchers working in these and related fields, and apply kCARTA to quantify how different spectroscopic databases impact simulated clear-sky Top of Atmosphere (TOA) brightness temperatures. Focusing on the infrared (605 - 2830 cm$^{-1}$) region, this paper begins with a description of the line-by-line code and the kCompressed database, followed by a description of the clear sky radiative transfer algorithm,

together with jacobians. The paper then discusses in detail some of the internal machinery of kCARTA, such as a background thermal computation developed for kCARTA, flux computations and scattering packages.

## 2 Overview of line-by-line code and kCompressed Database

### 2.1 UMBC-LBL

For an input set of [average temperature, pressure and gas amount (in molecules/cm$^2$)] parameters, a custom monochromatic

line-by-line code (UMBC-LBL) (De Souza-Machado et al., 2002) has been developed in order to accurately compute optical depths. This code defaults to the Van Vleck and Huber lineshape (Van Vleck and Huber, 1977; Clough et al., 1980) for almost all molecules, using spectroscopic line parameters from the HIgh-resolution TRANsmission (HITRAN) molecular absorption database.

For each spectral region the UMBC-LBL optical depth computations are divided into bins that are typically 1 cm$^{-1}$ wide in

the infrared. The optical depth in each of these bins is accumulated in three stages as shown in Figure 1 : (1) fine mesh stage - absorption due to line centers within 1 cm$^{-1}$ of the bin edges is included at a very high resolution (0.0005 cm$^{-1}$) and then five point boxcar integrated to the output 0.0025 cm$^{-1}$ grid; in Figure 1 these are the red lines within the bin edges at ± 0.5 cm$^{-1}$ and the blue lines within 1 cm$^{-1}$ of the same bin edges (2) medium mesh stage - absorption from line centers within 1-2 cm$^{-1}$ of the bin edges is included at 0.1 cm$^{-1}$ resolution, shown in green in the figure and finally (3) coarse mesh stage - absorption

from line centers within 2-25 cm$^{-1}$ of the bin edges, are included at 0.5 cm$^{-1}$ resolution (none shown in the figure); for (2) and (3) the results are interpolated to the output 0.0025 cm$^{-1}$ grid. The black line is the accumulated optical depth for that bin.

We note two points here. First, the above computations are very similar to those in other models (Edwards, 1992; Dudhia, 2017), but we use the "medium" bins and "coarse" bins for the lines whose centers are within the intervals lying ± [1,2] and ± [2,25] cm-1 respectively of the bin edges, instead of using only "coarse" bins. Second, we note that for most Earth atmosphere

molecules, the line strength × gas amount combination means the optical depth contribution due to line centers further than 25 cm$^{-1}$ away from the bin is negligible and can be ignored (Dudhia, 2017); the exception for the Earth atmosphere are H$_2$O and CO$_2$ which have countless strong lines further than 25 cm$^{-1}$ away from bin edges. To speed up the optical depth calculations, the weak but non-negligible contribution from these "far lines" is added in using a continuum optical depth contribution which depends on temperature and gas absorber amount.





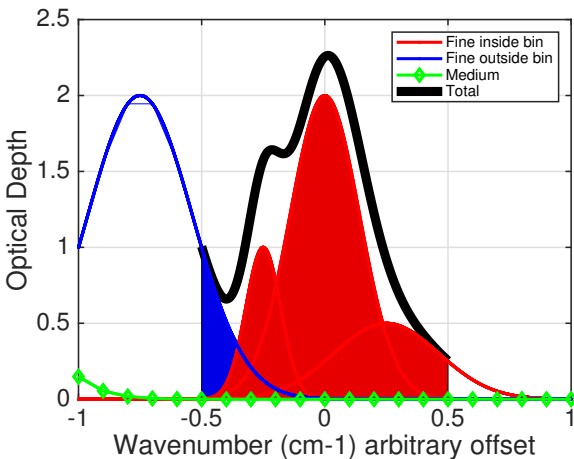

**Figure 1.** Line-by-line calculations from UMBC-LBL. The bin of interest is at (-0.5,+0.5) cm$^{-1}$. Lineshapes whose line centers are within this bin (red) and within $\pm$ 1cm$^{-1}$ of the bin edge (blue) are computed using high spectral resolution; line centers that are further out (green) have the lineshapes computed at medium resolution and then interpolated to a higher resolution; line centers even further away (not shown) are computed at coarse resolution. The black curve is the sum over all the line contributions within that bin.

The above steps are followed for almost all molecules. Modifications to the above steps are needed for water vapor (which is separated into the traditional "basement" plus "continuum" contributions (Clough et al., 1980, 1989)) and $CO_2$ in the 4 and 15 $\mu m$ region which needs line-mixing lineshapes (Strow and Pine, 1988; Tobin et al., 1996; Niro et al., 2005; Lamouroux et al., 2015). Other molecules have optical depths that are more easily modeled with the Van Huber lineshape, though recently the infrared absorption due to $CH_4$ has been modeled using line mixing (Tran et al., 2006). The UMBC-LBL optical depth

computation for water vapor should be robust at all frequencies, and allows the addition of water continuum models such as the recent MT-CKD 3.2 coefficients (Mlawer et al., 2012). Spectra from UMBC-LBL have been extensively compared against optical depths computed by models such as Line-by-line Radiative Transfer Model (LBLRTM) (Clough et al., 1992, 2005) and the General line-by-line Atmospheric Transmittance and Radiance model (GENLN2) (Edwards, 1992). For example the optical depths of $O_3$ in the 10 $\mu m$ region more closely resemble the reference monochromatic code MonoRTM than the LBLRTM code,

which uses an accelerated algorithm for calculating the Voigt function (Mlawer, 2016).

### 2.2   kCompressed database

When applied toward any realistic Earth atmosphere simulation for an observing instrument, the UMBC-LBL calculations described above become impractically slow as they need to be performed for multiple gases in the atmosphere, over $\sim$ 100 atmospheric layers and encompassing a wide spectral range.

UMBC-LBL is therefore primarily used to generate an uncompressed database of look-up tables as described below. For each gas other than water vapor, the spectra are computed using the US Standard Atmosphere temperature profile, as well as



five temperature offsets (in steps of 10K) on either side of the temperature profile, for a total of 11 temperatures. The default infrared database spans 605 cm$^{-1}$ to 2830 cm$^{-1}$, broken up into intervals that are 25 cm$^{-1}$ wide. Each file contain matrices to compute optical depths for 10000 points at 0.0025 cm$^{-1}$ point spacing. The one hundred average pressure layers used in
making the database are from the AIRS Fast Forward Model. The layers span 1100 mb to 0.005 mb (about ground level to 85 km), and were chosen such that there is less that 0.1 K Brightness Temperature (BT) errors in the simulated AIRS radiances. The layers are about 200 m thick at the bottom of the atmosphere, gradually getting thicker with height (about 0.65 km at 10 km and 6 km at an altitude of 80 km).

These $10000 \times 100 \times 11$ optical depths intervals are then compressed using Singular Value Decomposition (SVD) to produce
the kCompressed Database. Each compressed file will have a matrix of basis vectors $B$ (size $10000 \times N$), and compressed optical depths $D'$ (size $N \times 100 \times 11$), where $N$ is the number of significant singular vectors found. The prime denotes the compression worked more efficiently when the optical depths were scaled to the (1/4) power (Strow et al., 1998; Rodgers, 2000).

The self broadening of water is accounted for by generating monochromatic lookup tables for the reference water amount,
multiplied by 0.1, 1.0, 3.3, 6.7 and 10.0 at the eleven temperature profiles specified above, meaning $D'$ for water will have an extra dimension of length 5. Note that for the infrared we treat the HDO isotope (HITRAN isotope 4) as a separate gas from the rest of the water vapor isotopes.

The compressed optical depths $D'$ vary smoothly in pressure, meaning the user is not limited to only using the 100 AIRS layers. For an arbitrary pressure layering, the look-up tables are uncompressed using spline or linear interpolation in temperature
and pressure, and scaled in gas absorber amount. Temperature interpolation of matrix $D'$ for an AIRS 100 layer atmosphere therefore results in a matrix $D''$ of size $N \times 100$, and the final optical depths (of size $10000 \times 100$) are computed using $(BD'')^4$. Both the spline and linear interpolations allow easy computation of the analytic temperature derivatives, from which kCARTA can rapidly compute analytic Jacobians (see Section 6). The cumulative optical depth for each layer in the atmosphere is obtained by a weighted sum of the individual gas optical depths, with accuracy limited by that of the compressed database (Strow
et al., 1998). The interested reader is referred to (Vincent and Dudhia, 2017) for a further discussion of other RTAs that use compressed databases.

The most recent kCompressed database uses line parameters from the HITRAN 2016 database (Rothman et al., 2013; Gordon et al., 2017), which together with the UMBC-LBL lineshape models, determine the accuracy of the spectral optical depths in this database. UMBC-LBL $CO_2$ line-mixing calculations use parameters that were derived a few years ago. Newer line-mixing
models exist and we now use optical depths computed using LBLRTM v12.8 together with the line parameter database file based on HITRAN 2012 (aer_v_3.6), and (a) $CO_2$ line mixing by (Lamouroux et al., 2010, 2015)) and (b) $CH_4$ line mixing by (Tran et al., 2006).

In addition complete kCompressed databases for the IR using optical depths only from HITRAN 2012, LBLRTM v12.4 code and from GEISA 2015 (Husson et al., 2015) have been generated for comparison purposes. At compile time we usually point
kCARTA to the HITRAN 2016 kCompressed database made by UMBC-LBL, but at run time we have switches that easily allow us to swap in for example the $CO_2$ and $CH_4$ tables generated from LBLRTM.





The original lookup tables for the thermal infrared occupy hundreds of gigabytes, while the compressed monochromatic absorption coefficients are a much more manageable 824 megabytes (218 megabytes (water+HDO) + 76 megabytes (CO2) + 530 megabytes (about 40 other molecular and 30 cross section gases)). A general overview of some of the factors involved in compressing look-up tables for use in speeding up line-by-line codes is found in (Vincent and Dudhia, 2017), while more details about the detailed testing and generation of the kCARTA SVD compressed database are found in (Strow et al., 1998). Appendix B discusses the extension of the database to span $15\,\mathrm{cm}^{-1}$ to $44000\,\mathrm{cm}^{-1}$, though we note that kCARTA lacks built-in accurate scattering calculations in the shorter wavelengths. In order to resolve the narrow doppler lines at the top-of-atmosphere, the resolution of the spectral regions in Appendix B is adjusted according to $\delta\nu \simeq \nu_0\sqrt{(T/m)}$.

The default kCARTA mode is to use all 42 molecular gases in the HITRAN database, together with about 30 cross-section gases. If the user does not provide the profiles for any of these gases, kCARTA uses the US Standard profile for that gas. The user can also choose to only use a selected number of specified gases. While running kCARTA, the user can then define different sets of mixed paths, where some of the gases are either turned off or the entire profile is multiplied by a constant number, which is very useful when for example we want to include only certain gases when we parametrize optical depths for SARTA.

## 3 kCARTA Clear sky radiative transfer algorithm

As a stream of radiation propagates through a layer, the change in diffuse beam intensity $R(\nu)$ in a plane parallel medium is given by the standard Schwartschild equation (Liou, 1980; Goody and Yung, 1989; Edwards, 1992)

$$\mu\frac{dR(\nu)}{dk_e} = -R(\nu) + J(\nu) \tag{1}$$

where $\mu$ is the cosine of the viewing angle, $k_e$ is the extinction optical depth, $\nu$ is the wavenumber and $J(\nu)$ is the source function. For a non-scattering "clear sky", the source function is usually the Planck emission $B(\nu, T)$ at the layer temperature $T$, leading to an equation that can easily be solved for an individual layer. The general solution for a downlooking instrument measuring radiating propagting up through a clear-sky atmosphere can be written in terms of four components :

$$R(\nu) = R_{\mathrm{s}}(\nu) + R_{\mathrm{lay}}(\nu) + R_{\mathrm{th}}(\nu) + R_{\mathrm{solar}}(\nu) \tag{2}$$

which are the surface, layer emissions, downward thermal and solar terms respectively. In terms of integrals the expressions can be written as (see e.g. (Liou, 1980; Dudhia, 2017))

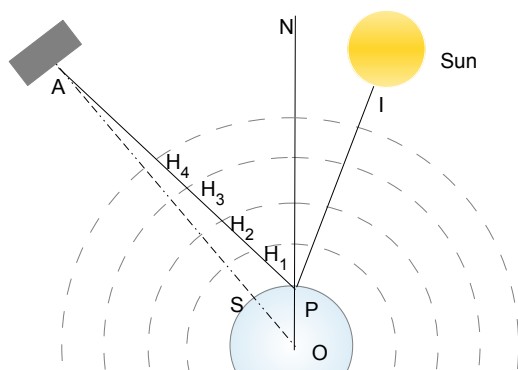

**Figure 2.** Viewing geometry for the sounders modeled by kCARTA. *A* is the satellite and point *P* is being observed by the satellite, while *I* is the sun

$$R(\nu,\theta) = \epsilon_s(\nu)B(\nu,T_s)\tau_{atm}(\nu,\theta) + \int\limits_{surface}^{TOA} B(\nu,T(z))\frac{\partial\tau(\nu,\theta)}{\partial s}ds +$$

$$\frac{1-\epsilon_s(\nu)}{\pi}\tau_{atm}(\nu)\int d\Omega \int\limits_{TOA}^{surface} B(\nu,T(s))\frac{\partial\tau(\Omega)}{\partial s}ds +$$

$$\rho_s(\nu)B_{\odot}(\nu)cos(\theta_{\odot})\tau_{atm}(\nu,\theta_{\odot})\tau_{atm}(\nu,\theta) \tag{3}$$

170    where $B(\nu,T)$ is the Planck radiance at temperature $T$, $T_s$ is the skin surface temperature, $\epsilon_s, \rho_s$ are the surface emissivity and reflectivity; $B_{\odot}(\nu)$ is the solar radiance at TOA, $\theta_{\odot}$ is the solar zenith angle; $\theta$ is the satellite viewing angle, , $\tau_i(\nu,\theta)$ is the transmission at angle $\theta$ while $\tau_{atm}$ is the total atmospheric transmission.

In what follows we discretize Eqn. 2 so that layer $i = 1$ is the bottom and $i = N$ (=100) the uppermost, schematically shown in Figure 2 for a clear sky four layer atmosphere, with *O* being the center of the Earth. *A* is the satellite while *S* is the satellite

175    sub-point directly below it. Point *P* is the ground scene being observed by the satellite (slightly away from nadir), and *N* is the local normal at *P*. ∠SAP is the satellite scan angle while ∠APN is the satellite zenith angle $\theta$; ∠NPI is the solar zenith angle $\theta_{\odot}$. Note that as the radiation propagates through the pressure layers from *P* to $H_1$ to $H_2$ to $H_3$ to $H_4$ to *A*, the local angle (between the radiation ray and the local normal at any of the concentric circles) keeps changing due to the spherical geometry of the layers (refraction effects can also be included).

180    The default mode of kCARTA assumes no variation of layer temperature with optical depth, uses a background thermal diffusive angle that varies with the layer-to-ground optical depth (instead of a constant value typically assumed to be $cos^{-1}(3/5)$) and does ray-tracing to account for the spherical atmospheric layers (but with no density effects). The f90 version of kCARTA



allows the layer temperature to vary linearly with optical depth, and to choose alternate ways of computing the background term, which will be discussed in Section 7. The individual contributions to the upwelling radiance are computed as follows.

## 3.1 Surface emission

kCARTA requires the user to supply the spectrally varying emissivity (and surface reflectance). The kCARTA surface emission is given by

$$R_s(\nu) \quad = \quad \epsilon_s B(\nu, T_s) \tau_{1 \to N}(\nu, \theta) \tag{4}$$

## 3.2 Layer emission

The atmospheric absorption and re-emission is modeled as :

$$R_{\text{lay}} \quad = \quad \sum_{i=1}^{i=N} B(\nu, T_i)(1.0 - \tau_i(\nu))\tau_{i+1 \to N}(\nu, \theta) \tag{5}$$

Layers with negligible absorption ($\tau_i \to 1$) contribute negligibly to the overall radiance, while those with large optical depths ($\tau_i \to 0$) "black" out radiation from below. $(1.0 - \tau_i(\nu))$ is the emissivity of the layer while $(1.0 - \tau_i(\nu))\tau_{i+1 \to N}(\nu, \theta)$ is the weighting function $W_i$ of the layer.

## 3.3 Solar radiation

Letting the surface reflectance be denoted by $\rho_s(\nu, \theta, \phi)$, then the solar contribution to the TOA radiance is given by

$$R_{\odot}(\nu) \quad = \quad \rho_s(\nu, \theta, \phi) B_{\odot}(\nu) cos(\theta_{\odot}) \times \tau_{N \to ground}(\nu, \theta_{\odot})) \tau_{ground \to N}(\nu, \theta)) \Omega_{\odot} \tag{6}$$

Over ocean, if the wind speed and solar and satellite azimuth angles are known, the reflectance can be pre-computed using the Bi-directional Reflectance Distribution Function (BRDF) and input to kCARTA; see for example Appendix C in (Nalli et al., 2016). It is not easy to compute the BRDF over land, and the reflectance could be simply modeled as $\rho_s(\nu) = \frac{1 - \epsilon_s(\nu)}{\pi}$.

$\Omega_{\odot} = \pi(r_s/d_{se})^2$ is the solid angle subtended at the earth by the sun, where $r_e$ is the radius of the sun and $d_{se}$ is the earth-sun distance. The solar radiation incident at the TOA $B_{\odot}(\nu)$ comes from data files related to the ATMOS mission (Farmer et al., 1987; Farmer and Norton, 1989), and is modulated by the angle the sun makes with the vertical, $cos(\theta_{\odot})$ (day-of-year effects are not included in the earth-sun distance).

## 3.4 Background thermal radiation

The atmosphere also emits radiation downward, *at all angles*, in a manner analogous to the upward layer emission just discussed. The total background thermal radiance at the surface is an integral over all (zenith and azimuth) radiance streams





propagating from the top-of-atmosphere (set to 2.7 K) to surface. This is time consuming to compute using quadrature, and one approximation is to use a single effective (or diffusivity) angle of $\theta_{diff} = cos^{-1}(3/5)$ at all layers and wavenumbers :

$$R_{th}^{surface}(\nu) \quad = \quad \pi\rho_s \sum_{i=N}^{i=1} B(T_i) \left[ \tau_{i-1 \rightarrow ground}(\nu, \theta_{diff}) - \tau_{i \rightarrow ground}(\nu, \theta_{diff}) \right] \qquad (7)$$

The summation is from top-of-atmosphere to ground, and $\rho_s$ is the surface reflectance discussed above. Current sounders have channel radiance accuracy better than 0.2K, so while the above term is much smaller than the surface or upwelling atmospheric emission contributions, it has to be computed accurately. Section 7 includes a detailed discussion of how kCARTA improves the accuracy of this background term by using a look-up table to rapidly compute a spectrally and layer varying
diffusive angle.

## 4    Impact of spectroscopy on TOA BTs

In this section kCARTA is used to evaluate how spectroscopy impacts the simulated BTs at the TOA. First we evaluate differences between spectral databases (HITRAN 2012, HITRAN 2016 and GEISA 2015), and then we assess how spectroscopic line parameter uncertainties impact the BTs. The satellite scan angle in all calculations is 22°, which is about half the maximum
scan of the AIRS, CrIS and IASI sounders.

### 4.1    Top of Atmosphere BTs computed using different databases

For this section three spectral databases were used to compute TOA radiances (converted to BT) using kCARTA. A set of 49 regression profiles typically expected in the Earth atmosphere (Strow et al., 2003) were used throughout this paper for this and other studies. The profiles in this set include the US Standard temperature, pressure, and trace gas constituent fields
(McClatchey et al., 1972), as well as the Mid-Latitude Summer/Winter, Polar Summer/Winter, Tropical profiles, and a set of extreme and intermediate hot/cold/dry/humid profiles chosen from the Thermodynamic Initial Guess Retrieval (TIGR) database (Achard, 1991) to span the expected variablilty of profiles in the Earth's atmosphere (Strow et al., 2003). The results are shown in Figure 3.

For the left panels (a) three databases were used : (1) default HITRAN 2016 (2) HITRAN 2012 and (3) GEISA 2015; all three
used $CO_2$,$CH_4$ limemixing from LBLRTM v12.8 and we set to zero all optical depths from cross-section gases. For clarity we show results for a simulated AIRS instrument by convolving the computed monochromatic radiances using the AIRS Spectral Response Functions; the gaps in the plots are where AIRS has no detectors. This exercise is purely to show differences in the spectroscopy manifesting as equivalent BT differences, with validation against actual clear-sky observations left for a future paper. Panel (a) of Figure 3 show the mean (top) and standard deviation (bottom) of the differences between HITRAN 2016
and [GEISA 2015 (blue), HITRAN 2012 (red)]; the black curves are the AIRS NeDT. The obvious spectral differences are in the regions of the $H_2O$ lines, especially in the 6.7 $\mu m$ region; there are also noticeable differences in the HITRAN 2012 10 $\mu m$ $O_3$; differences are also seen in the 2400 - 2700 $cm^{-1}$ region. These spectral differences are to be expected from the slightly





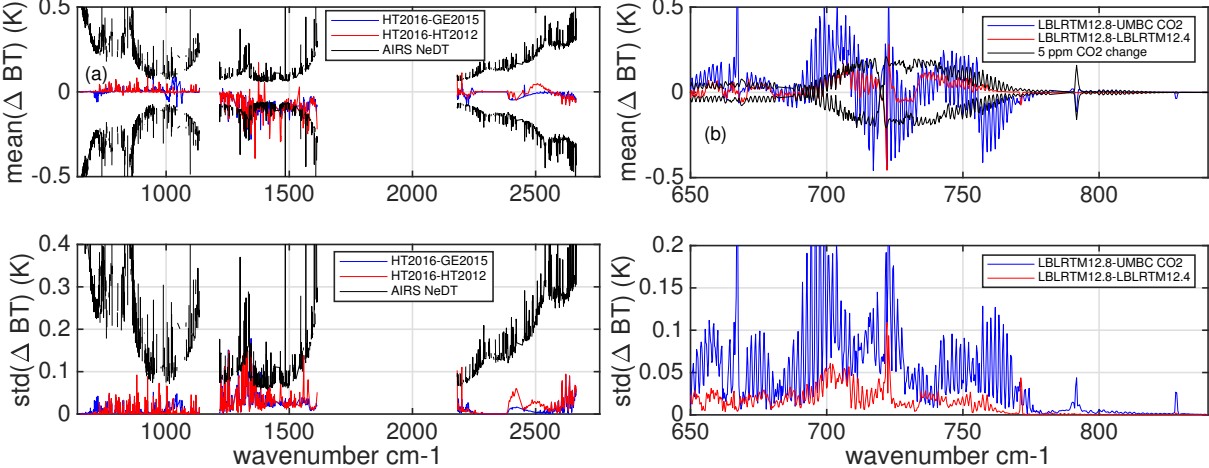

**Figure 3.** Comparing different spectroscopic databases, with top/bottom panels showing the mean and standard deviation over 49 regression profiles. Left panel (a) : H2016 versus GEISA 2015 (blue) and H2012 (red). The black curve is the AIRS NeDT. Right panel (b) : 15 $\mu m$ CO2 line mixing differences, compared to LBLRTM 12.4 (red) and UMBC-LBL (blue). For both sets of plots the top is the mean differences versus the control (HITRAN 2016 with $CO_2$,$CH_4$ line mixing from LBLRTM v12.8, evaluated over our 49 regression profile set

differing spectroscopy in the individual databases; the interested reader who wishes to more completely fully interpret them is referred to the papers describing these individual databases.

The right panels (b) of Figure 3 shows the mean (top) and standard deviation (bottom) of the differences between the $CO_2$ spectroscopy between default LBLRTM v12.8 and two other databases : (1) UMBC-LBL (blue) and (2) LBLRTMv12.4 (red)]. The black curves in the top panel are the column $CO_2$ jacobian for a $\pm$ 5 ppm change, and shows the differences in the LBLRTM $CO_2$ versions are within that amount; the bottom sagain shows the standard deviation of the differences. The differences compared to the older UMBC-LBL line mixing model (which was developed using HITRAN 2000 line parameters) is much larger in the 690-720 $cm^{-1}$ region. A perusal of the HITRAN 2000 and 2012 line parameters for the main $CO_2$ isotope

shows the main updates to the spectroscopy are to the air- and self-broadening linewidth coefficients in the 600-730 $cm^{-1}$ range, which would impact the line mixing relaxation matrices. We note the SARTA model currently in use for AIRS L2 retrievals was developed in 2010 and has been tuned against observations; a more updated model using HITRAN 2016 line parameters together with LBLRTM 12.8 $CO_2$,$CH_4$ line mixing spectroscopy is currently being developed.

**4.2 Impact of spectroscopy error budget on Top of Atmosphere BT**

The HITRAN and GEISA databases are bundled with error indices associated with the individual spectroscopic line parameters. The uncertainty codes in the HITRAN database (Rothman et al., 2005) are replicated in Table 4.2 for convenience. As seen from the table, the uncertainties are divided into two groups : absolute (wavenumber and pressure shift) and relative (intensity and broadening parameters).





| | Wavenumber and Pressure shift (cm-1) | | Intensity, Halfwidths, and Temperature-dependence |
|---|---|---|---|
| Code | Uncertainty Range | Code | Uncertainty Range |
| 0 | $\geq 1.$ or Unreported | 0 | Unreported or Unavailable |
| 1 | $\geq 0.1$ and $< 1.$ | 1 | Default or Constant |
| 2 | $\geq 0.01$ and $< 0.1$ | 2 | Average or Estimate |
| 3 | $\geq 0.001$ and $< 0.01$ | 3 | $\geq 20\%$ |
| 4 | $\geq 0.0001$ and $< 0.001$ | 4 | $\geq 10\%$ and $< 20\%$ |
| 5 | $\geq 0.00001$ and $< 0.0001$ | 5 | $\geq 5\%$ and $< 10\%$ |
| 6 | $\geq 0.000001$ and $< 0.00001$ | 6 | $\geq 2\%$ and $< 5\%$ |
| 7 | $\geq 0.0000001$ and $< 0.000001$ | 7 | $\geq 1\%$ and $< 2\%$ |
| 8 | $\geq 0.00000001$ and $< 0.0000001$ | 8 | $< 1\%$ |
| 9 | $\geq$ Better than 0.00000001 | | |

**Table 1.** Meaning of HITRAN uncertainty codes

It is relatively straightforward to use these indices to include the associated uncertainty of any relevant line parameter for most gases, and generate a new compressed database using UMBC-LBL. Exceptions arise because the HITRAN 1986 database edition did not have uncertainties and was populated with zeros, a few of which have not yet been updated (Gordon, 2018). In these cases the uncertainty index is 0 (for the left column of the table) or 0,1,2,3 (for the right part of the table). The value of 0 occurs for example, in many of the line strength uncertainties for the 10 $\mu m$ $O_3$ isotopes 1,2; to remedy this we used 3% which

is slightly lower than the 4% estimated in (Drouin et al., 2017; Birk et al., 2019). Similarly many of the strong 7.6 $\mu m$ $CH_4$ lines (isotope 1) are assigned an intensity uncertainty code of 3; to our knowledge there is no other additional information and we used a maximum value of 20% (Gordon, 2018). We also used 0.1 cm$^{-1}$, 0.1 cm$^{-1}$/atm uncertainties for the line center and pressure shift when the uncertainty index was 0(Gordon, 2018).

    In Figures 4 and 5 below, we concentrate on the changes in computed BT effects after we perturb individual parameters for

the first few molecules in the HITRAN database ($H_2O$,$O_3$,$N_2O$,$CO$,$CH_4$). The red curves in the top and bottom panels show mean and standard deviations of the $\Delta$(BT) averaged over 49 regression profiles. The black curve are the AIRS NeDT at 250 K. We avoid perturbing $CO_2$, as it would involve coupling to the LBLRTM line mixing code (with its associated parametrized continuum); to a certain extent the right hand panel of Figure 3 alleviates this omission as it compares different linemixing models. We used the Van Huber lineshape for $CH_4$ to avoid these same line-mixing complexities.

The perturbed parameters are line strength and all the broadening parameters (Figure 4), and wavenumber and line center shift due to pressure (Figure 5). These figures show that the uncertainty in the HITRAN line parameters are typically less than the noise of the current generation of sounding instruments; the reality is the biggest problems are older $O_3$ and $CH_4$ strengths and broadening uncertainty indices that need to be updated (Gordon, 2018). A similar calculation where we assumed the uncertainties were independent of each other allowed us to randomize all perturbations to be within 0 and $\pm X$ (where $X$

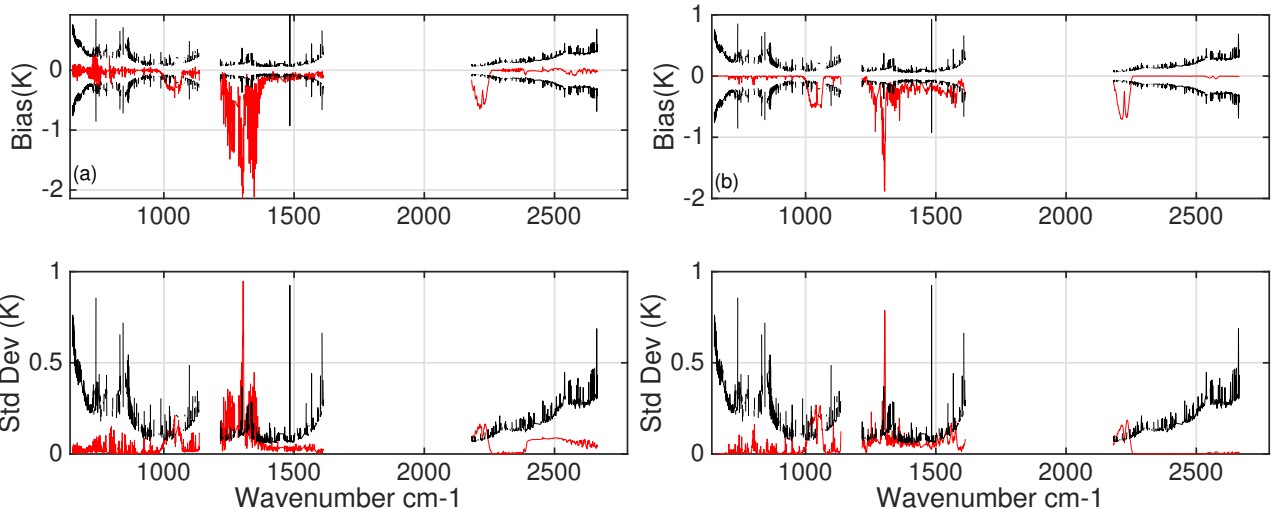

**Figure 4.** Δ(BT) for perturbations to (a) strength and (b) broadening parameters. The top and bottom panels show mean and standard deviations of the changes, while the black curves are the AIRS NeDT at 250 K. Many of the $CH_4$ line uncertainties (near the 1305 cm$^{-1}$ region) were set at 20% because of the lack of information for those lines.

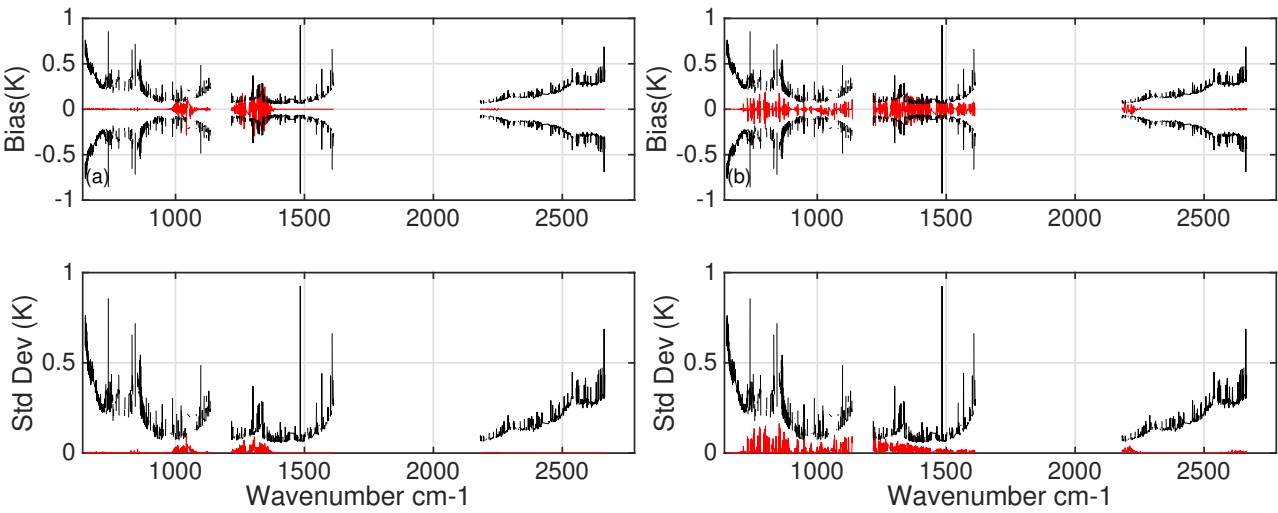

**Figure 5.** Δ(BT) for perturbations to (a) wavenumber and (b) line shift due to pressure parameters. The top and bottom panels show mean and standard deviations of the changes, while the black curves are the AIRS NeDT at 250 K. Most of the perturbations were below the 0.1 cm$^{-1}$ level.





is the maximum number corresponding to the error indices in Table 4.2); this roughly halves the largest errors shown in these
two figures, so they are within $\pm$ 1 K in the $CH_4$ region.

## 5    Non Local Thermodynamic Equilibrium computations

During the daytime, incident solar radiation is preferentially absorbed by some $CO_2$ and $O_3$ infrared bands, whose kinetic
temperature then differs from the rest of the bands or molecules. This leads to enhanced emission by the lines in these bands.
Limb sounders detect NLTE effects in the 15 $\mu m$ $CO_2$ bands (and in other molecular bands for example $O_3$) due to the
extremely long paths involved, but these are not modeled in the package as kCARTA is designed for nadir sounders.

For a nadir sounder, the most important effects are seen in the $CO_2$ 4 $\mu m$ ($\nu_3$) band. kCARTA includes a computationally
intensive line-by-line Non Local Thermodynamic Equilibrium (NLTE) model to calculate the effects for this $CO_2$ band. The
model requires the kinetic temperature profile and NLTE vibrational temperatures of the strong bands in this region, to compute
the optical depths and Planck modifiers for the strong NLTE bands and the weaker LTE bands (Edwards et al., 1993, 1998;
Lopez-Puertas and Taylor, 2001; Zorn et al., 2002), which are then used to compute a monochromatic top-of-atmosphere nadir
radiance.

AIRS provided the first high resolution nadir data of NLTE in the 4 micron $CO_2$ band. Using the kCARTA NLTE line-by-line
model, a Fast NLTE Model (De Souza-Machado et al., 2007) for sounders has already been developed, which is used in the
NASA AIRS L2 operational product.

## 6    Clear Sky Jacobian algorithm

Retrievals of atmospheric profiles (temperature, humidity and trace gases) minimize the differences between observations and
calculations, by adjusting the profiles using the linear derivatives (or jacobians) of the radiance with respect to the atmospheric
parameters. This section describes the computation of analytic jacobians by kCARTA. For a downward looking instrument,
for simplicity consider only the upwelling terms in the radiance equation (atmospheric layer emission and the surface terms).
Assuming a nadir satellite viewing angle, the solution to Equation 1 is:

$$R(\nu) = \epsilon_s B(T_s, \nu) \tau_{1 \to N}(\nu) + \Sigma_{i=1}^{i=N} B(T_i, \nu)(1.0 - \tau_i(\nu)) \tau_{i+1 \to N}(\nu) \qquad (8)$$

Differentiation with respect to the $m$-layer variable $s_m$, (gas amount or layer temperature $s_m = q_{m(g)}, T_m$) yields

$$\frac{\partial R(\nu)}{\partial s_m} = \epsilon_s B(T_s) \frac{\partial \tau_{1 \to N}(\nu)}{\partial s_m} + \sum_{i=1} B(T_i, \nu)(1.0 - \tau_i(\nu)) \frac{\partial \tau_{i+1}(\nu)}{\partial s_m} + \sum_{i=1} \tau_{i+1 \to N} \frac{\partial}{\partial s_m} [B(T_i, \nu)(1.0 - \tau_i(\nu))] \qquad (9)$$





where as usual, $\tau_m(\nu) = e^{-k_m(\nu)}$, $\tau_{m \to N}(\nu) = \Pi_{j=m}^{N} e^{-k_j(\nu)}$. The differentiation yields

$$
\begin{aligned}
\frac{\partial R(\nu)}{\partial s_m} = & \left[\epsilon_s B(T_s)\tau_{1 \to N}\right](-1)\frac{\partial k_m(\nu)}{\partial s_m} + \\
& \left[\sum_{i=1}^{m-1}(1.0 - \tau_i(\nu))B_i(\nu)\tau(\nu)_{i+1 \to N}\right](-1)\frac{\partial k_m(\nu)}{\partial s_m} + \\
& \left[(1.0 - \tau_m(\nu))\frac{\partial B_m(\nu)}{\partial s_m} - B(T_m, \nu)\frac{\partial \tau_m(\nu)}{\partial s_m}\right]\tau_{m+1 \to N}(\nu)
\end{aligned}
\tag{10}
$$

The individual Jacobian terms $\frac{\partial k_m}{\partial s_{m(g)}}$ are rapidly computed by kCARTA, as follows. The gas amount derivative is sim-

ply $\frac{\partial k_m}{\partial q_{m(g)}} = \frac{k_m}{q_{m(g)}}$ (with added complexity for water, to account for self broadening), and the temperature derivative $\frac{\partial k_m}{\partial T}$ is cumulatively obtained $while$ kCARTA is performing the temperature interpolations during the individual gas database uncompression.

The solar and background thermal contributions are also included in the Jacobian calculations. The thermal background Jacobians are computed at $\cos^{-1}(3/5)$ at $all$ levels, for speed. This would lead to slight differences when comparing the

Jacobians computed as above to those obtained using finite differences. The Jacobians with respect to the surface temperature and surface emissivity are also computed, as are the weighting functions.

## 7   Background thermal and temperature variation in a layer

In this section we take a closer look at the computation of downwelling background thermal radiation, and layer temperature variation.

### 7.1   Background thermal radiation

The contribution of downwelling background thermal to top-of-atmosphere upwelling radiances is negligible in regions that are blacked out as the instrument cannot see surface leaving emission. Similarly in layers/spectral regions where there is very little absorption and re-emission, the contribution is negligible as the effective layer emissivity (denoted by $\Delta\tau_i(\nu)$ below) goes to zero. The background contribution thus needs to be done most accurately in the window regions (low but finite optical

depths) ; depending on the surface reflectance in the window regions, in terms of BT this term contribute as much as 4 K of the total radiance when reflected back up to the top of the atmosphere.

The contribution at the surface by a downwelling radiance stream propagating at angle $(\theta, \phi)$ through layer $i$, is given by

$$
\begin{aligned}
\Delta R_i(\nu, \theta, \phi) &= B(\nu, T_i)(1.0 - \tau_i(\nu, \theta, \phi))\tau_{i-1 \to ground}(\nu, \theta, \phi) \\
&= B(\nu, T_i)(\tau_{i-1 \to ground}(\nu, \theta, \phi) - \tau_{i \to ground}(\nu, \theta, \phi))
\end{aligned}
\tag{11}
$$





where $\theta$ is the zenith and $\phi$ is the azimuth angle, and $\tau_{i\rightarrow ground}$ are the layer-to-ground transmittances, derived from layer-to-ground optical depths $x$. This equation can be rewritten as

$$\Delta R_i(\nu,\theta,\phi) = B(\nu,T_i) \times \Delta\tau_i(\nu,\theta,\phi) \qquad (12)$$

An integral over $(\theta,\phi)$ would give the contribution from the layer. The total downwelling spectral radiance at the surface would be a sum over all $i$ layers (and the downwelling flux at the surface would be the integral over all wavenumbers).

The integral over the azimuth is straightforward (assuming isotropic radiation), but the integral over the zenith is more complex. Since the reflected background term is much smaller than the surface or atmospheric terms, a single stream at the effective angle $\theta_{diff} = cos^{-1}(3/5)$ (Liou, 1980) is often used as an approximation, at all layers and wavenumbers.

We have refined the computation as follows. Recall that $\Delta R(\nu)$ in Equation 12 depends on the layer-to-ground optical depth $x$. Letting $\mu = cos\theta$ the integral over the zenith ($\int_0^1 \mu d\mu e^{-x/\mu} = E_3(x)$. The area under the $E_3(x)$ curve would be the total

flux coming all optical depths ($0 \le x \le \infty$); over 77% of this area comes from the range $0 \le x \le 1$.

Applying the Mean Value Theorem for Integrals (MVTI) to $E_3(x)$, we can write Eq. 12 in terms of two effective diffusive angles $\theta_d^i, \theta_d^{i-1}$ at each layer $i$ :

$$\Delta\tau(i,i-1) = \tau(i-1 \rightarrow ground, \theta_d^{i-1}, \nu) - \tau(i \rightarrow ground, \theta_d^i, \nu)$$

$$R_{th}^{surface}(\nu) = 2\pi\rho_s \sum_{i=N}^{i=1} B(\nu,T_i)\Delta\tau(i,i-1) \qquad (13)$$

with the effective angles varying as a function of the layer to ground space optical depth of that layer, and the layer immediately below it. Numerical solutions to the MVTI show that when $x \rightarrow 0$ then $\mu_d \rightarrow 0.5$ (or $\theta_d \rightarrow 60°$). Similarly as $x \rightarrow \infty$ then $\theta_d \rightarrow 0°$, but this optically thick atmosphere means an instrument observing from the TOA cannot see the surface, so we use a lower limit (of 30°) for the diffusive angle. Finally when $x = 1.00$ we find the special case $\mu_d = 0.59274 \simeq (3/5)$. For "optically thin" regions, the layers closest to the ground contribute most to $R_{th}(\nu)$.

With today's high speed computers, kCARTA uses an effective diffusive angle $\theta_d$ tabulated as a function of layer to ground optical depth $x$, as follows. For each 25 cm$^{-1}$ interval spanning the infared the layer $L$ above which $cos^{-1}(3/5)$ can be safely used was determined; below this layer, the lookup table is used. The table has higher resolution for $x \le 0.1$ and becomes more coarse as $x$ increases, with the effective diffusive angle cutoff at 30° when the optical depths are larger than about 15.

We have tested this method of computing the background thermal against both 20 point Gauss-Legendre quadrature and the
3 point exponential Gauss-quadrature (used by LBLRTM flux computations), and found the method to very accurate and fast, both in terms of the downwelling flux at the surface, and also the final TOA computed radiance, even when the emissivity is as low as 0.8 (which means a significant contribution from the reflected thermal). At this low emissivity value, the constant $acos(3/5)$ diffusivity angle model produces final TOA BT which differ from the Gauss-Legendre model by as much as 1.3 K (for the tropical profile) at for example 900 cm$^{-1}$, while the exponential quadrature and our model have errors smaller than
0.005 K.





## 7.2 Variation of layer temperature with optical depth

LBLRTM (Clough et al., 1992, 2005) has been extensively tested and shown to be very accurate, in its computation of optical depths, radiances and fluxes. In the computation of radiances, the main difference between the kCARTA and LBLRTM codes is that for each spectral point, the former defaults to a "constant in $\tau$" layer temperature variation, while the latter uses a "linear in $\tau$" layer temperature variation. Here we summarize the relevant equations, and briefly discuss the computed differences using kCARTA. For an individual layer, with lower and upper boundary temperatures $T_L, T_U$, the "linear in $\tau$" approximation leads to the following expression for the radiance at the top of the layer (re-written from Equation 13 in (Clough et al., 1992))

$$I(\nu) = I_0(\nu)T + (1-T)\left\{B_{av}(\nu) + (B_u(\nu) - B_{av}(\nu))\left(1 - 2\left(\frac{1}{\tau} - \frac{T}{1-T}\right)\right)\right\} \tag{14}$$

where the optical depth $\tau$ includes the view angle $\tau = \tau_{layer}/cos(\theta)$ and transmission $T = exp(-\tau)$. $I_0(\nu)$ is the radiation incident at the bottom of the layer, $B_{av}(\nu)$ is the Planck radiance corresponding to the average layer temperature, while $B_u(\nu)$ is the Planck radiance corresponding to the upper boundary. For large $\tau$, $T \to 0$ and $I(\nu) \to B_u(\nu)$. For small $\tau \to 0$ the expression can be further expanded as follows

$$I(\nu) = I_0(\nu)T + (1-T)\left\{B_{av}(\nu) + (B_u(\nu) - B_{av}(\nu))\left(\frac{\tau}{6} - \frac{\tau^3}{360} + \frac{\tau^5}{15120}\right)\right\} \tag{15}$$

Comparing to the top of layer radiance in the "constant in $\tau$" model,

$$I(\nu) = I_0(\nu)T + (1-T)B_{av}(\nu) \tag{16}$$

one sees the expressions are identical if there is no temperature variation ie $(B_u(\nu) = B_{av}(\nu))$. The default kCARTA model layers are approximately 0.25 km thick (or a temperature spread of about 1.5 K for a 6K/km lapse rate) at the bottom of the atmosphere, and about 2 km thick in the stratosphere (a temperature difference of 10 K). However the gaseous absorption in these upper layers is typically negligible, except deep inside the strongly absorbing 15 $\mu m$ and 4 $\mu m$ $CO_2$ bands. Differences in BTs computed using the "constant" versus "linear" models will be expected to be greatest in these regions. The US Standard, Tropical, Mid Latitude and Polar Summer/Winter profiles were used to evaluate the differences between the "constant" versus "linear" models. The monochromatic results clearly showed the expected mean BTDs as large as 10 K or more in the optically thick $CO_2$ (15$\mu m$ and 4 $\mu m$) regions, and systematically much smaller in the window or regions dominated by $O_3$ and $H_2O$ absorption. After convolving the computed radiances over the AIRS response functions, the mean differences typically dropped to below 0.05 K in the opaque regions (*except* exactly at the peak of the dominant 667 cm$^{-1}$ Q-branch where the mean difference was about 0.2 K).





## 8 Flux Computations

Longwave fluxes at the top and bottom of the atmosphere, as well as the heating and cooling rates are computed by integrating spectral radiances from Equation 2 over all angles, and over the infrared spectral region : kCARTA is limited to the spectral

range 15 - 3000 $cm^{-1}$ spanned by the different bands of kCARTA (see Appendix B). The major limitation of kCARTA for flux calculations is the fixed 0.0025 $cm^{-1}$ spectral (infrared) resolution at every layer, compared to the varying-with-height resolution employed by other models such as LBLRTM. This impacts the high altitude longwave cooling in the 15 $\mu m$ $CO_2$ band.

We use the Rapid Radiative Transfer Model (RRTM-LW) (Mlawer et al., 2012) as our reference model for flux and heating

rate comparisons in a clear sky atmosphere. This fast model compute fluxes and heating rates in 16 bands spanning 10 $cm^{-1}$ to 3000 $cm^{-1}$, and was developed using LBLRTM; the latter uses a varying spectral resolution at each layer ($\delta\nu$ equal to 4 points per half-width in each layer) which means the spectra for the upper atmosphere layers have very high resolution. kCARTA uses the same approach as RRTM and LBLRTM to compute fluxes and heating rates : the angular integration uses an exponential Gauss-Legendre with 3 or 4 terms, with a "linear in $\tau$" layer temperature variation.

The accuracy of the flux and heating rate algorithm in kCARTA at the default 0.0025 $cm^{-1}$ resolution was assessed by comparing fluxes and heating rates in the dominant 15 $\mu m$ to 10 $\mu m$ bands (fourth to eighth RRTM bands, spanning 630-1180 $cm^{-1}$) computed using RRTM and kCARTA, using the 49 regression profile set.

At 0.0025 $cm^{-1}$ resolution the kCARTA and RRTM-LW heating rates differ by less than 0.2 K/day on average for altitudes below 40 km, but at higher altitudes the differences were much larger, and could be 1.5 K/day. This was attributed to the default

spectral resolution of kCARTA in the 15 $\mu m$ region. To test this, we generated a database of resolution 0.0005 $cm^{-1}$ spanning 605-1205 $cm^{-1}$ for $H_2O$, $CO_2$ and $O_3$ which are by far the dominant absorbers in this spectral region, especially at higher altitudes. This significantly improved the results, with heating rate differences dropping to about 0.2 K/day almost everywhere.

Figure 6 shows the heating rate differences between kCARTA and RRTM. The left panel shows differences between kCARTA and RRTM, with the mean and standard deviation being solid and dashed respectively; the right panel shows mean calculations

as a function of height. The blue curves were done at default 0.0025 $cm^{-1}$ resolution while the red curves were done at higher 0.0005 $cm^{-1}$ resolution. While the agreement is better than 0.05 K/day in the lowest 30 km, Figure 6 shows the heating rates using the low resolution begin to differ noticeably above 45 km (blue curve); conversely the high resolution heating rates (red curves) are within 0.2 K/day till about 65 km.

Based on the above, users interested in flux calculations can improve the accuracy while retaining the speed advantages of

kCARTA by dividing the thermal infrared into two regions : 605-805 $cm^{-1}$ at 0.0005 $cm^{-1}$ resolution while the remaining 805-2830 $cm^{-1}$ region can have the current 0.0025 cm-1 resolution.

## 9 Scattering package included with f90 kCARTA

The daily coverage of hyperspectral sounders provides us with information pertaining to the effects of cloud contamination on measured radiances. Ignoring these effects can negatively impact retrievals used for weather forecasting and climate modeling.

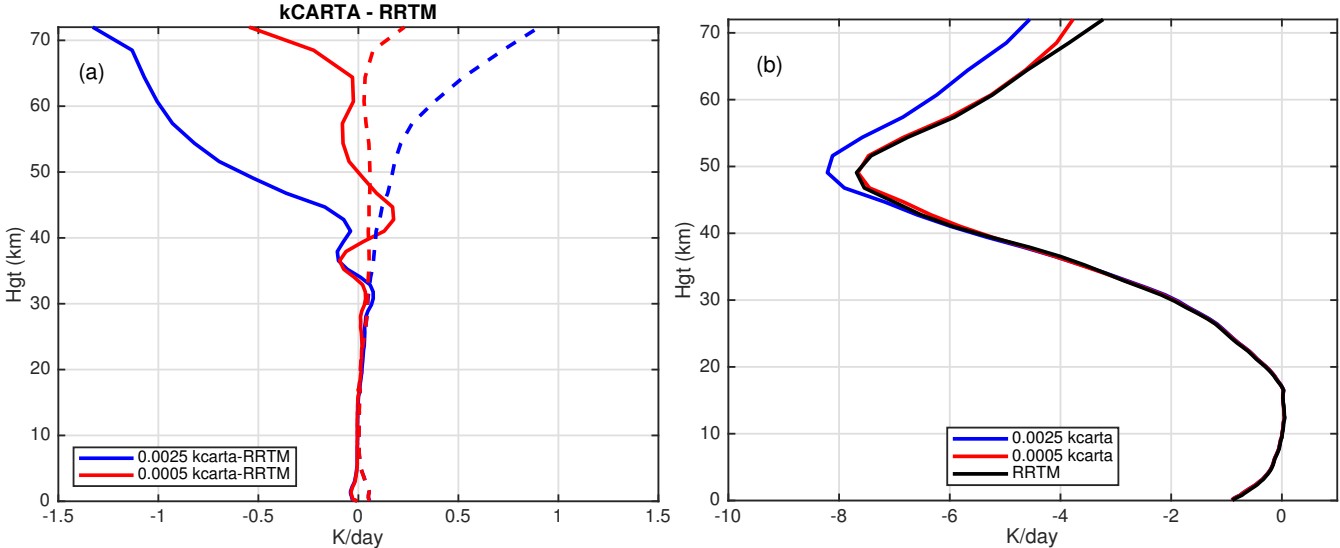

**Figure 6.** Heating rates for the 630-1180 cm$^{-1}$ region computed using low (0.0025 cm$^{-1}$; blue curves) and high (0.005 cm$^{-1}$; red curves) resolution kCARTA databases for 49 spanning Earth atmosphere profiles, compared to RRTM. (a) The left panel shows differences in the heating rates, while (b) the right panel shows the mean heating rates.

A scattering package based on the PCLSAM (Parametrization of Cloud Longwave Scattering for use in Atmospheric Models) scheme (Chou et al., 1999) has been interfaced into f90 kCARTA (see Appendix C). The implementation allows kCARTA to compute radiances very quickly in the presence of scattering media such as clouds or aerosol. For a given scattering species and assumed particle shape and distribution, the extinction coefficients, single scattering albedo and asymmetry parameters needed by the scattering code are stored in tables as a function of wavenumber and effective particle size (for a particle amount

of 1 $g/m^2$). In addition one can easily use kCARTA to output monochromatic optical depths that can be imported into well known scattering packages. More details about PCLSAM and our cloud representation models are found in Appendix C.

## 10 Conclusions

We have described the details of a very fast and accurate pseudo- monochromatic code, optimized for the thermal infrared spectral region used by operational weather sounders for thermodynamic retrievals. It is much faster than line-by-line codes,

and the accuracy of its spectroscopic database has been extensively compared against GENLN2 and more recently to LBLRTM. Updating the spectroscopy in a selected wavenumber region for a specified gas is as simple as updating the relevant file(s) in the database : for example, our custom UMBC-LBL enables us to re-build entire databases within weeks of the latest HITRAN release.

The computed clear sky radiances includes a fast, accurate estimate of the background thermal radiation. Analytic tempera-

ture and gas amount Jacobians can be rapidly computed. Early in the AIRS mission, comparisons of AIRS observations against





kCARTA simulations allowed for the quick implementation of modifications to gas optical depths : our modifications to the CKD2.4 and MT-CKD 1.0 continuum versions are very similar to what is now in the MT-CKD2.5 version. We now use the MT-CKD 3.2 continuum, together with the $N_2$ and $O_2$ continuum contributions bundled with that same version. The default infrared spectral resolution of 0.0025 cm$^{-1}$ is good enough for nadir hyperspectral sounders; however at this resolution kCARTA has inaccuracies modeling fluxes in the $CO_2$ 15 $\mu m$ band and radiances from limb sounders whose weighting function peak high in the atmosphere, though one can switch to a higher resolution database if needed.

We assessed how the uncertainties in the line parameters of the HITRAN database impact top of atmosphere BTs for a nadir sounder, and found it typically is less than the NeDT of the current generation of sounders. The exceptions are $O_3$ and $CH_4$, but the remedy likely lies in correcting the uncertainty indices rather than the determination of the parameters themselves. In the future we plan to assess the $CO_2$ line-mixing package supplied with the HITRAN package, and plan to evaluate the impact of line parameter uncertainties on TOA BTs at that point.

kCARTA is fast enough to be used in Optimal Estimation retrievals for instruments spanning a reasonably small wavenumber range. The kCARTA database has been extended to include 15-44000 cm$^{-1}$, which eventually needs to be updated to HITRAN 2016 (see the Appendix). In the future we plan to augment the optical depth calculations performed by UMBC-LBL by using speed-dependent lineshapes as parameters become available.

## Appendix A: Code availability

The UMBC-LBL code has been developed in Matlab, with extensive use of Mex files to speed up loops. The package is available at https://github.com/ sergio66/umbc-lbl-spectra, and is fully described in (De Souza-Machado et al., 2002).

The f90 and Matlab versions of kCARTA can be cloned from https://github.com/ sergio66/kcarta_gen and https://github.com/strow/kcarta-matlab respectively.

## A1   UMBC-LBL and kCARTA downloads and auxiliary requirements

The kCompressed Database is supplied in Fortran little-endian binary files that contain the optical depths for a specific gas. Each file contains optical depths at 10000 spectral points × average pressures corresponding to the 100 AIRS layers. Links to the (605 - 2830 cm$^{-1}$) compressed database can be found at http://asl.umbc.edu/pub/packages/kCompressedDatabase.html

We also supply the US Standard Profile for all gases in the database, and kLAYERS, a program that takes in a point profile (from sondes and NWPs) and outputs an AIRS 100 layer path averaged profile (in molecules/$cm^2$). kLAYERS needs our supplied HDF file implementation (RTP) source code.

The Matlab version should work with R2012+ while the compiler for the fortran version must support structures, such as Absoft, ifort and PGF. As the RTP file contains the atmospheric profile and scan geometry, both the Matlab and f90 kCARTA only need a simple additional (namelist) file to drive either code. The f90 version of kCARTA outputs binary files, which typically have header information such as kCARTA version number, number of layers and gases and parameter setting values, followed by panels, each 10000 points long, containing the optical depths, radiances, jacobians or fluxes computed and output





by kCARTA. A number of Matlab based readers can then be used to further process the kCARTA output as needed. More information is found at http://asl.umbc.edu/pub/packages/ kcarta.html.

**Appendix B:  Available spectral regions and f90 kCARTA features**

**Table B1.** Spectral bands for kCARTA

| Band (cm$^{-1}$) | Point Spacing (cm$^{-1}$) | Band center ($\mu m$) | number of files |
|---|---|---|---|
| 15-30 | 0.00005 | 444.4 | 30 |
| 30-50 | 0.00010 | 250.0 | 20 |
| 50-80 | 0.00015 | 153.8 | 20 |
| 80-140 | 0.00025 | 90.9 | 28 |
| 140-300 | 0.00050 | 45.4 | 34 |
| 300-500 | 0.00100 | 25.0 | 21 |
| 500-605 | 0.00150 | 18.1 | 07 |
| 605-2830* | 0.00250 | 5.8 | 89 |
| 2830-3550 | 0.00250 | 3.1 | 30 |
| 3550-5550 | 0.00100 | 2.2 | 21 |
| 5550-8250 | 0.01500 | 1.4 | 19 |
| 8250-12000 | 0.02500 | 0.98 | 16 |
| 12000-25000 | 0.05000 | 0.54 | 26 |
| 25000-44000 | 0.10000 | 0.29 | 19 |

The UMBC-LBL line-by-line code has been used to generate optical depths in the spectral regions seen in Table B1. The 605-2830 cm$^{-1}$ band is marked with an asterisk, since our work focuses on this spectral region. The current database in this spectral region uses lineshape parameters from HITRAN 2016. The Van Vleck and Huber lineshape is used for all HITRAN molecules from ozone onwards; water vapor uses the "without basement" plus MT-CKD 3.2, and $CO_2$,$CH_4$ use line-mixing

optical depths generated from LBLRTM v12.8. Note that in the important 4.3 $\mu m$ temperature sounding region, the f90 version can also include the $N_2$/$H_2O$ and $N_2$/$CO_2$ Collision Induced Absorption (CIA) effects modeled in (Hartmann et al., 2018; Tran et al., 2018), which depend on $CO_2$,$H_2O$ and $N_2$ absorber amounts.

A clear-sky radiance calculation in the infrared takes about 20 seconds, using a 2.8 GHz 32 core multi-threading Intel machine. The run-time goes to 120 seconds if Jacobians are also computed (for 9 gases). A full radiance calculation from 15

to 44000 cm$^{-1}$ takes less than 5 minutes.

Table B2 lists a number of the features of kCARTA, with the ones marked by a asterisk only available in the f90 version. Note that the tables defaults to describing the spectroscopy for the infrared region.





Table B2: kCARTA features; * indicates currently only available in the f90 version

| Feature | Default | Options |
|---|---|---|
| SPECTROSCOPY | | |
| (1) IR Database | HITRAN 2016 | HITRAN 2012, GEISA 2015 |
| (2) Resolution | 0.0025 cm$^{-1}$ | can make other resolutions |
| (3) Molecular gases | HITRAN ID 1-42 | choose some |
| (4) Cross-section gases | HITRAN (CFCs etc) | choose some |
| (5) Water continuum | MT-CKD 3.2 | e.g. MT-CKD 1.0, 2.5 |
| (6) $CO_2$ line mixing | from LBLRTM v12.8 | UMBC-LBL line mixing |
| (7) $CO_2$/$H_2O$, $CO_2$/$N_2$ CIA* | off | Hartmann and Tran* |
| (8) $CH_4$ line mixing | from LBLRTM v12.8 | None (voigt) |
| (9) $O_2$ and $N_2$ | HITRAN 2016 and | |
| | LBLRTM v1.28 continuum | |
| (10) NLTE | line by line * | SARTA approximation |
| (11) Uncompressiom | linear | spline |
| RADIATIVE TRANSFER | CLEAR SKY | |
| (1)Temperature variation | Constant | Linear in $\tau$* |
| (2) Background thermal | acos(3/5) in upper layers, | acos(3/5) all layers |
| | accurate angle lower layers | gauss quadrature* |
| (3) Background thermal | lambertian | |
| Surface reflection | $(1 - \epsilon_s(\nu))/\pi$ | |
| (4) Solar reflection | user specified | |
| (5) Ray tracing | Spherical atmosphere,$n$=1 | $n$ varies* |
| (6) Direction | downwelling | upwelling |
| (7) Solar | from tables | use 5600 K |
| (8) Jacobians | 100 layer $T, WV$, | column jacs |
| | weighting functions | |
| | acos(3/5) backgrnd thermal | |
| (9) Fluxes* | upwell, downwell | Heating rates |
| RADIATIVE TRANSFER | ALL SKY | |
| (1) TwoSlab Cloud model* | PCLSAM | fluxes and jacobians |
| (2) couple to LBLDIS* | | |



## Appendix C: PCLSAM scattering algorithm

The *PCLSAM* scattering algorithm for longwave radiances has applications ranging from dust retrievals (De Souza-Machado et al., 2010), to modeling the effects of clouds on sounder data (Matricardi, 2005; Vidot et al., 2015). This scattering model changes the extinction optical depth from $k(\nu)$ to a parametrized number $k^{scatterer}_{eff.extinction}(\nu)$ (Chou et al., 1999), and is designed for cases of the single scattering albedo $\omega$ being much less than 1, such as in the thermal infrared, where $\omega$ for cirrus and water droplets and aerosols is typically on the order of 0.5.

Since $k^{scatterer}_{eff.extinction}(\nu)$ is now effectively the absorption due to the cloud or aerosol, for each layer $i$ that contains scatterers we replace the gas absorption optical depth with the total absorption optical depth

$$k_{total}(\nu) = k^{gases}_{atm}(\nu) + k^{scatterer}_{eff.extinction}(\nu) \tag{C1}$$

where(Chou et al., 1999) $k^{scatterer}_{eff.extinction}(\nu) = k^{scatterer}_{extinction}(\nu) \times (1 - \omega(\nu))(1 - b(\nu)))$ and the backscatter $b(\nu) = (1 - g(\nu))/2$. Using this for every layer containing scatterers, the radiative transfer algorithm is now the same as clear sky radiative transfer,
with very little speed penalty.

    kCARTA is capable of using a TwoSlab (De Souza-Machado et al., 2018) cloud representation scheme for use with *PCLSAM*. This allows for non unity fractions for up to two clouds, so that radiative transfer then assumes the total radiance is a sum of four radiance streams (clear, cloudy 1, cloud 2 and the cloud overlap) weighted appropriately :

$$r(\nu) \quad = \quad c_{overlap}r^{(12)}(\nu) + c_1 r^{(1)}(\nu) + c_2 r^{(2)}(\nu) + f_{clr}r^{clr}(\nu) \tag{C2}$$

With this model kCARTA allows the user to specify upto two types of scatterers in the atmosphere (ice/water, ice/dust, water/dust or even ice/ice, water/water, dust/dust); the two scatterers are placed in separate "slabs" which occupy complete AIRS layers and are specified by cloud top/bottom pressure (in millibars), cloud amount (in g/m2), cloud effective particle diameter (in $\mu m$). After the computations are done, all five radiances are output when two clouds are defined (overlap, two clouds separately, clear, and the weighted sum), and three radiances if only one cloud is defined (one cloud, clear, weighted
sum).

    Analytic jacobians for temperature, gas amounts, and cloud micro-physical parameters (effective size and loading) can also be computed, as can be fluxes and associated heating rates, though the slab boundaries could introduce spikes in the heating rate profiles.

    Matlab routines read in kCARTA optical depths output and break them into separate files, which can be piped into LBLDIS
(Turner et al., 2003; Turner, 2005), a code that merges optical depths and scattering using the extensively tested Discrete Ordinates Radiative Transfer (DISORT) (Stamnes et al., 1988) algorithm.



**Author contribution**

Sergio DeSouza-Machado prepared the manuscript with contributions from all (living) co-authors. The initial compressed database coding and testing was done by L. Strow, H. Motteler and S. Hannon. Following this deS-M wrote the Fortan and Matlab wrapper codes for clear sky radiative transfer and jacobians, which were tested and validated by the other authors. Scattering and flux capabilities were added and tested by deS-M.

**Competing interests**

The authors declare that they have no conflict of interest.

**Acknowledgments**

This work was supported in part by NASA grant number NNG04GG03G-2. Dave Tobin of UW-Madison helped with UMBC-LBL $CO_2$ line-mixing code and modifying the water continuum coefficients. Dave Edwards of NCAR provided the GENLN2 line-by-line code to compare kCARTA against. Both Dave Edwards and Manuel Lopez-Puertas of the Instituto de Astrofisica de Andalucia (Spain) contributed to the NLTE portions of the code. Optical depth and flux comparisons against LBLRTM were facilitated by Eli Mlawer (Atmospheric and Environmental Research, Lexington MA). The impact of spectroscopic uncertainty to BTs at the TOA were carried out partly for the ITOVS Radiative Transfer Working Group, of which Marco Matricardi (ECMWF) is the leader : Iouli Gordon (Harvard-Smithsonian) suggested how to tweak the uncertainties when the indices were 0,1,2, as did Brian Drouin (JPL) for ozone line strengths.





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
