# Peer review of "kCARTA: A fast pseudo line-by-line radiative transfer algorithm with analytic jacobians, fluxes, Non-Local Thermodynamic Equilibrium and scattering for the infrared"

_Atmospheric Measurement Techniques, 2019_

## Referee Comment (RC1) · Anonymous Referee #1 · 16 Sep 2019

kCARTA: A fast pseudo line-by-line radiative transfer algorithm with analytic Jacobians, fluxes, Non-Local Thermodynamic Equilibrium and scattering for the infrared

by S. DeSouza-Machado et al. August 2019.

Referee comments.

OVERVIEW

The paper describes an 'intermediate' radiative transfer model kCARTA: faster than the line-by-line model (eg UMBC-LBL) but slower than the models used in operational processing of data from the new generation of hyperspectral instruments (eg SARTA). As a monochromatic model, this has the advantage of using Beer's Law to combine transmittances but relies on pre-computed absorption coefficients at a fixed set of pressure, temperature and (for H2O) mixing ratio points, which introduces a (small) interpolation error compared with line-by-line models.

With increasing spectral resolution of instruments and increased computing power, monochromatic models such as kCARTA may soon form the basis of operational retrieval or assimilation schemes (eg the OSS model), so this represents an important development and the algorithms should be documented in a journal such as AMT. However, there are several major issues with the current paper that I feel should be addressed.

GENERAL COMMENTS

1) There is no comparison of kCARTA radiances with a more well-established (and preferably line-by-line) model, not even the source models UMBC-LBL or LBLRTM. The kCompressed tables have been previously verified to reproduce the absoption coefficients or optical paths, but this paper deals with all the extra components of a radiative transfer model.

2) I have concerns about the impact of the coarse spectral resolution: 0.0025cm-1. Where does this figure come from? The usual requirement for radiative transfer is to be able to capture the signal from Doppler-broadened lines in the upper stratosphere, which have typical mid infrared widths of 0.001 cm-1, hence resolutions of 0.0005 or 0.001 cm-1 are generally considered necessary. Given the inherent flexibility of the kCompressed tables, I am also surprised that the authors have not considered using an adaptive, rather than fixed, spectral grid, so that the spectral resolution is concentrated around the line-centres, so obtaining better accuracy for the same size compressed datasets and computation time.

3) I am unclear on the conversion of atmospheric profile quantities (temperature, pres-

sure, composition) to the (presumably?) equivalent homogeneous paths represented by the kCompressed data. Are these absorber-weighted equivalent (ie Curtis-Godson) temperatures and pressures? Or simply layer means? Given the use of a spherical atmosphere (what radius of curvature is assumed?) rather than a plane-parallel assumption, presumably some sort of numerical integration scheme is required to obtain these quantities, even the total amount of absorber in a layer.

4) Section 4, on the impact of spectroscopy on TOA radiances, seems an unnecessary digression. While spectroscopic uncertainties are certainly an issue that merits attention, that's not really anything to do with the kCARTA model being introduced (besides which all the variation is handled in the generation of the kCompressed datasets, which are to some extent independent). The data gap in the plots, arising from the gap in AIRS coverage, is also undesirably for such a comparison. It would have been more useful to see comparisons of kCARTA TOA radiances against other models instead.

5) For most molecules, kCARTA uses data created by the UMBC-LBL model using the Van Vleck and Huber lineshape. Since the Voigt lineshape is very much the 'standard', there should be some explanation of how this differs and why it is used in preference to Voigt.

6) The inclusion of non-LTE effects, just for the $CO_2$ 4um band, seems to require the inclusion of a separate line-by-line model within kCARTA. This is a huge overhead in complexity for a relatively specialised application. Given the kCARTA structure, it seems a more natural approach would have been to incorporate vibrational temperature as an extra axis on the kCompressed datasets, and just use GENLN2 to calculate these. Alternatively, if you are including a LBL model within kCARTA, at least extend it so that LBL (with or withouth non-LTE) can be used for any molecule.

7) The calculation of background thermal radiation (section 7) uses an interesting idea, and would probably have merited some expansion as a separate (if somewhat specialised) article by itself. I have a number of questions, detailed in the minor comments

below.

TYPOS/MINOR COMMENTS

P1 L20: '... data are presently ...'

P1 L20: Given Susskind was describing cloud clearing for the previous generation of meterological instruments I'm not sure it's an appropriate reference for the hyperspectral instruments.

P1 L20: The text may be read as implying that there is some sort of correction applied to the spectra to remove the influence of clouds, whereas I suspect it is more accurate to say that any cloud-contaminated spectra are simply rejected.

P2 L27: What determines this 0.0025cm-1 figure? As a rough estimate I expect it would be determined by the requirement to resolve the Doppler widths of lines, and given molecular velocities are of the order of c/10ˆ6 that would correspond to around 0.001 cm-1 at 1000 cm-1.

P2 L27: I am not convinced by the argument that the computation of optical depths at high spectral resolution for 50 or so profiles for the training set is something that needs accelerating. Surely this is something that only has to be done once and, even if occasionally repeated, the fact that it takes a week rather than a couple of hours isn't really be an issue.

P2 L40: I assume the issue with replacing line-by-line with kCompressed data is one of the accuracy of the absorption coefficient or reconstructed layer optical depths. However to state that the 'radiances' are accurately reproduced requires a whole new set of tests to verify the accuracy of the radiative transfer through an atmosphere.

P2 L49: Missing second ')' after 1999.

P3 L71: I have not come across the Van Vleck and Huber lineshape. Is this different to the more conventional Voigt? If so, how, and why the unconventional choice?

Fig 1: I appreciate this is just a sketch, but the black Total line doesn't seem to be a sum of the red, blue and green lines. Also raise the '-1' in the x-axis title to a superscript (and in subsequent figures).

P4 L98: Regarding MonoRTM - what is the point being made here?

P5 L107: +/- 50K does not seem a large range in temperature. Do you have any evidence that it spans the full range of atmospheric variability?

P5 L108: 'contains'

P5 L122: HITRAN 2016 lists two further isotopologues for water vapour containing a single deuterium atom and either a 17O or an 18O oxygen isotope (these are HITRAN isotopologues 5 and 6). Are these included with HDO or with the remaining H2O isotopologues?

P6 L144: It is not clear why the cross-section molecules are also represented using kCompressed databases. Presumably these end up much larger than the original files, which usually have only a few tens of (pressure,temperature) points and a much coarser spectral axis. Also using CIA probably won't work with LUTs - how will these new data be used?

P6 L150: HITRAN 2016 lists 49 rather than 42 molecules, and a number of these (or even 1:42) are not represented in the US Standard Atmosphere.

P6 L158: 'Schwarzschild'

P7 Eq(3): The solid angle integration should just be over a hemisphere and should include the B.dt/ds term scaled by cos(e) where e is the elevation angle 0:pi/2 in the hemispheric integration (thus the integral of cos(e).dOmega from 0:2pi on its own should yield pi).

P7 Eq(3): I don't see why the cos(theta_sun) appears in the last term on its own, but it seems there should be some solid angle integration over the sun's disk (as in Eq 6)

otherwise it will be as if the whole sky radiates at the solar temperature.

P7 L170: Isn't 1-epsilon_s(nu) the same as rho_s(nu)?

P7 L171: extra comma near end of line

P7 L171: There is no tau_i term in Eq(3), just tau and tau_atm.

P7 L180: Assuming the temperature profile is specifed at points P,H1,H2 etc what constant temperature is assumed for, eg the lowest layer? Is it T(P), T(H1), or something else?

P7 L182: By 'density effects' do you mean refraction?

P7 L186: Both emissivity *and* reflectance have to be supplied? Eq(4) only uses emissivity.

P8 Eq(5): The indexing doesn't seem to work. Interpreting tau_(i+1 to N) as the transmittance from the base of layer i+1 to the base of layer N the calculation for layer 3 in the diagram would be ( 1 - tau_3 ).tau_(4 to N) but here N=4 so tau_(4 to N) = 1 whereas it should be the transmittance through layer 4. Similarly Eqs 8-10

P8 L195 Better to swap sections 3.3 and 3.4 to match the same order these terms appear in Eq (3)

P8 L196 rho is now defined as reflectance, but for Eq 3 it was reflectivity. Is there a difference?

P9 L213 As a general comment, it would be nice to have a plot of the magnitude of these four terms as a function of the infrared spectrum, assuming say some fixed surface emissivity of around 0.98 (so 2% diffuse reflectance).

P9 Section 4: Presumably for this exercise different sets of kCompressed databases were computed by running UMBC-LBL etc for the different sets of spectroscopic data, and then running kCARTA using these 3 different sets of kCompressed databases.

If one really wants to demonstrate the differences in TOA radiance couldn't one simply run the LBL models with the different spectroscopic data and eliminate the whole intermediate step of generating kCompressed datasets?

P9 L230 'linemixing'

P10 L243 'sagain' ?

P10 L247 it is not clear that differences have anything at all to do with linemixing - it seems they might simply reflect differences in the standard line widths that would be evident whether line-mixing effects were included or not?

P13 Section 5: It seems odd that kCARTA has a non-LTE line-by-line module - it is something I would have expected in the UMBC-LBL code. Have you considered modelling non-LTE using kCompressed datasets? That would seem more in keeping with the overall design. Perhaps you would need an extra tabulated dimension in vibrational temperature, or (vib-kin) temperature?

P13 Eq 9: Summations should be from i=1,N. Also, tau_(i+1 to N) in second summation should have (\nu) afterwards.

P14 L304: 'Jacobian' from here to the end of the section start to be capitalised - inconsistent with earlier 'jacobian'

P14 L305: How is dB/dT calculated? (where s_m = T_m).

P14 L310: I don't understand the last sentence - what's the difference between the Jacobian and the weighting function wrt surface temperature and emissivity?

P14 L320: 'contributes'

P15 L334: the d\mu should come after the exp(-x/\mu).

P14 L334: Is there any significance in labelling this integral as E_3 ?

P14 L334: No closing bracket to match '('.

P14 Eq 13: From discussing downward radiation, the introduction of reflectance rho_s suggests to me that you are now modelling the reflected component of this radiance just above the surface. Where does the 2pi come from? I think you need to explain some of the intermediate steps.

P14 L340: If I understand this correctly, instead of approximating the downwelling radiance as if it comes from a fixed angle cos=3/5, you are adjusting the angle for each atmospheric layer according to the surface-layer optical thickness x.

It would be nice to see some plots of the comparisons with a full hemispheric integration to show that this is sigifnicantly more accurate than the fixed angle assumption. Also, is there a reason why the cut off at 30deg has to be applied for optically thick atmospheres? Otherwise it might also be useful for modelling the radiance viewed by upward-looking instruments, or downward looking instruments close to the surface.

P15 L346: Given the rapid spectral variation in any 25cm-1 interval, I don't understand how you can assign a single assumption to the whole interval. Won't there be a whole range of optical thickness within the 25cm-1 region so that the assumption works better for some spectral points than others?

P15 Sec 7.2: The linear-in-tau model, where optical depth is scaled by the sec(theta) to allow for off-nadir viewing angles, assumes a plane plane-parallel atmosphere where theta is fixed for the layer (and the same for every layer). How is this handled for the spherical atmospheres assumed in kCARTA where cos(theta) can vary significantly when viewing off-nadir?

P16 L365: It depends what you mean by the 'average layer temperature'. For the optically thin limit you would expect this to converge to the Curtis-Godson temperature, ie the absorber-weighted mean temperature, which would generally be at an altitude below the layer mid-point.

P16 L377: Even though it largely disappears after convolution (presumably because

the AIRS spectral resolution means that the radiance is dominated by contributions from lower altitudes), this 10 K difference does seem to be a serious issue. And one which would limit the use of kCARTA for accurately representing finer resolution instruments. Is this really due to not implementing linear in tau or could it be that kCARTA only uses a crude representation of layer temperature rather the something more physically justified such as Curtis Godson temperature? Why doesn't kCARTA just use the linear-in-tau model?

P17 L390 'computes'

P17 Flux Computations: are these with spherical or plane-parallel atmospheres?

P17 L407: Given the differences found from the linear-in-tau model in the previous section, could the differences in heating rates at high altitude simply be another manifestation of the same problem, ie assumption of constant temperature within thick layers at higher altitudes?

P18 Fig 6 caption: should be '0.0005' instead of '0.005'.

P18 L425: 'accuracy of its spectroscopic database'. I assume this refers to the kCompressed tables used as kCARTA input as opposed to the usual meaning which would be HITRAN or GEISA spectroscopic databases. But comparisons against GENLN2 or LBLRTM wouldn't just be a comparison of the kCompression with the original HITRAN unless your tests were for simple homogeneous paths where transmittance could be verified independently of other model assumptions such as ray-tracing and integration through atmospheric layers.

P19 L434: '0.0025 cm-1 is good enough for nadir hyperspectral sounders' - this is a contentious statement, and needs some justification (there is none in this paper).

P21 Table B2: lists (6) Direction as downwelling by default, upwelling as an option. Isn't it the other way around?

P22 L488: extra ')'.

P22 L495: 'up to'

——-

---

## Referee Comment (RC2) · Anonymous Referee #2 · 19 Oct 2019

This manuscript describes recent development of a pseudo-monochromatic radiative transfer package, kCARTA (the kCompressed Atmospheric Radiative Transfer Algorithm). The kCARTA improves computational speed relative to a line-by-line (LBL) radiative transfer model by using compressed optical depth lookup tables and by performing radiative transfer calculations at a pseudo-monochromatic grid. The main idea of the kCARTA algorithm is to compress molecular optical depth lookup tables into an Empirical Orthogonal Function (EOF) domain using a singular value decomposition method. The optical depth lookup tables, which are functions of atmospheric

molecules, wavelength (or frequencies), pressure, and temperatures, were generated using an UMBC-LBL model. Water vapor lookup table has an additional dependency on water vapor amount to account for self-broadening effect. The kCARTA approach reduces the optical depth lookup table size from hundreds of gigabytes to less than 900 megabytes. The generation of monochromatic transmittances from the compressed database is orders of magnitude faster than using a line by line model. The kCARTA package includes surface emission, atmospheric layer emission, solar reflection, and diffuse thermal emission reflected by the surface. The authors also described details on calculating non-local thermodynamic equilibrium in the CO2 4 micrometer band, the clear sky Jacobian calculation, a better approximation of calculating the background thermal radiations using an optical depth dependent single-stream effective diffusive angle, and the flux computation. The manuscript is well-written and the reviewer recommends publishing it with minor revisions. The specific comments are as follow:

1). What studies have the authors done to ensure that the 11 temperature grid points are adequate to represent the temperature dependency of optical depths of gases for each atmospheric layer. Please quantify the errors of interpolation due to 11 point grid and the choice of interpolation method (spline vs linear). 2). The kCARTA package is optimized for the thermal infrared spectral region. Though the authors claim it is trivial to extend the database out to span the far infrared to ultra-violet range, the package does not include Rayleigh scattering and an accurate multiple scattering radiative solver, which are important for the shortwave top of atmosphere radiation calculations. 3). On page 2 line 40, it will be useful to describe the relative errors between kCarta and the MNLBL. 4). Section 2.1 on page 4. Although the authors described the UMBC-LBL model line shape calculations and mentioned that extensive comparisons with LBLRTM and GENLIN2 have been performed, no quantitative results have been shown to illustrate differences among different LBL models. For some molecules, a sub-Lorentz line shape is used by LBLRTM, what about the UMBC-LBL? 5). The method used by kCARTA for calculating the downwelling background radiative is very efficient and much more accurate relative to a constant diffusive angle. However, for

a non-Lambertian surface (e.g a specular reflection surface), this may not be a good approximation.

There are some minor errors in the manuscript:

1. In the figure caption for Figure 6, 0.005 cm-1 should be 0.0005 cm-1. 2. Not all the symbols used in this paper are defined. For example, Omega in equation (3), g under equation (C1)... 3. Some of the links given in Appendix A are not available. 4. Line 149, the symbols in the equation are not explained by the text or the appendix B. 5. Line 230, please correct typo "limemixing"

---

## Referee Comment (RC3) · J.M. Blaisdell (Referee) · 24 Oct 2019

This paper documents a software package that is used by others, and therefore represents a useful contribution to the literature and is very appropriate for this journal. Since the software was used by the AIRS Science Team and others in product generation and will be used in future generation of products from hyperspectral infrared sounders, it is essential to get the details on the record. The authors describe their transmittance modeling with emphasis on approximations and limitations and highlight their innovations and the differences between their method and choices made by others

in the field. The paper is well written and I recommend its publication with only minor corrections. I have one substantive comment and identify a number of typographical errors.

Substantive comment: This paper describes a particular incarnation of kCARTA which the authors are making available. It would be very helpful to have a version identifier for the version described in the paper, and perhaps further identification, if possible, of which previous versions have been widely used. This is particularly needed in the paragraph beginning at Line 132, where the phrase "we now use" is unclear as to which previous version is obsolete.

Typographical errors:

Line 28, add comma after "process"

Line 59-60, mismatched parentheses

Line 158, "Schwarzschild" is misspelled

Line 163, "radiation propagating" are both misspelled

Line 171, remove the extra comma

Line 242, unmatched bracket

Line 243, "again" misspelled

Line 322, remove comma after "i"

Line 334, mismatched parentheses

Line 335, coming "from" all

Line 493, "cloudy" should be "cloud"

Line 536, "A.B." should be Boynard, A.

Line 536, "P.M." should be Pommier, M.

[Figure]

Line 536, "A.R." should be Razavi, A.

Line 537, "Chenm" should be "Chem"

Line 546, "Iancono" should be "Iacono"

Line 580, "humidity" misspelled

Line 584, list of authors is incomplete, at least add et al.

Line 613, "I.E. G." should be "Gordon, I.E."

Line 632, "Wm." should be removed

Line 649, "Karlsruhe", not "Karlsruhr"

Line 649, this link is dead. I find the paper at https://www.imk-asf.kit.edu/downloads/SAT/kopra_docu_part02.pdf

---

## Author Response (AR1)

Sergio DeSouza-Machado,
Department of Physics,
University of Maryland Baltimore County,
1000 Hilltop Circle,
Baltimore MD 21250

November 19, 2019

Dr. Lars Hoffmann,
Associate Editor,
Atmospheric Measurement Techniques

Dear Dr. Hoffmann,

This cover letter is included with the submission of our revised paper (amt-2019-282) entitled

"kCARTA : A fast pseudo line-by-line radiative transfer algorithm with analytic Jacobians, fluxes, Non-Local Thermodynamic Equilibrium and scattering for the infrared"

together with our responses to the reviewers.

In writing the manuscript, as before we have adhered as strictly as possible to the AMT manuscript guidelines. This document includes

- Page 1 : this cover letter,
- Page 2 : preamble for responses to reviewers,
- Pages 3-10: point-by-point response to Reviewer 1 (submitted earlier today as amt-2019-282-AC1-supplement.pdf),
- Page 11 : point-by-point response to Reviewer 2 (submitted earlier today as amt-2019-282-AC2-supplement.pdf),
- Pages 12-13 : point-by-point response to Reviewer 3 (submitted earlier today as amt-2019-282-AC3-supplement.pdf),
- Pages 14-15 : list of relevant changes from the above point-by-point responses,
- Pages 16-45 : output from latexdiff between the original and revised manuscripts

I have also separately uploaded the revised manuscript to AMT. We are confident we have taken great care to address all the concerns of the three referees, and look forward to your favorable response. Should you need to contact me, my email address is sergio@umbc.edu.

Sincerely,

Sergio De Souza-Machado

Authors responses to Atmos. Meas. Tech. Discuss., https://doi.org/10.5194/amt-2019-282

**kCARTA : A fast pseudo line-by-line radiative transfer algorithm with analytic Jacobians, fluxes, Non-Local Thermodynamic Equilibrium and scattering for the infrared**

by DeSouza-Machado et. al.

We thank the anonymous referees for their detailed read of the paper and providing introspective comments, many of which have resulted in changes to the revised version of the manuscript. In particular we have improved the accuracy of the computed kCARTA radiances by changing our default options (now linear-in-tau, higher resolution spectral database in the 15 um region). This update has already been pushed to github. We have also shortened the paper by removing the sections describing comparisons between the HITRAN/GEISA/$CO_2$ line-mixing databases, and the impact of spectroscopic uncertainties on TOA radiances. This has been replaced by a section where we compare kCARTA versus LBLRTM TOA radiances. Below we detail our responses to their individual concerns. For ease of review, we type-faced the reviewers questions in blue. When we refer to pages and line numbers in our answers, the context should make it clear whether we are talking about the original manuscript or our current revised manuscript.

Reviewer 1

*Specific comments*

1) There is no comparison of kCARTA radiances with a more well-established (and preferably line-by-line) model, not even the source models UMBC-LBL or LBLRTM. The kCompressed tables have been previously verified to reproduce the absorption coefficients or optical paths, but this paper deals with all the extra components of a radiative transfer model.
We have added a section that describes detailed inter-comparisons against LBLRTM.

2) I have concerns about the impact of the coarse spectral resolution: 0.0025cm-1. Where does this figure come from? The usual requirement for radiative transfer is to be able to capture the signal from Doppler-broadened lines in the upper stratosphere, which have typical mid infrared widths of 0.001 cm-1, hence resolutions of 0.0005 or 0.001 cm-1 are generally considered necessary. Given the inherent flexibility of the kCompressed tables, I am also surprised that the authors have not considered using an adaptive, rather than fixed, spectral grid, so that the spectral resolution is concentrated around the line-centers, so obtaining better accuracy for the same size compressed datasets and computation time.
As clearly stated in the title and abstract of the manuscript, kCARTA is a pseudo (monochromatic) line-by-line code, written specifically to compute radiances which are accurate when convolved with the Spectral Response Functions of the new generation of hyperspectral nadir sounders (or larger than 0.1 cm$^{-1}$ resolution). The loss of information due to compression with a finite number of basis vectors, means that for some molecules with complicated lineshapes we cannot completely reproduce the true monochromatic optical depths. However after convolution with a typical sounder response functions (resolution typically 0.5 cm$^{-1}$ at 15 $\mu m$), kCARTA is shown to easily compute accurate radiances (especially keeping in mind the large spectroscopic uncertainties that still exist in $CO_2$ line mixing, and larger detector NeDT at the long wavelengths, see for example Figs 3,4,5 of the original manuscript). We have re-written parts of the Paragraph 1, Page 2 to emphasize this.
We also thank the reviewer for pointing out we can easily improve kCARTA inter-comparisons against for example LBLRTM by changing the defaults to be (a) linear-in-tau and (b) higher resolution at the 15 $\mu m$ region (605-880 cm$^{-1}$). The above mentioned additional section will demonstrate we have tested and implemented this; any user interested is using higher resolutions can easily do so, by generating the appropriate database.
We have previously explored constructing and using databases of lower resolution in the lower atmosphere, and gradually increasing the resolution as the layer pressure decreases, but we found that while we could retain the TOA accuracy, the code was slowing down. More importantly the SVD compression works best with single resolution. The compressed files are small enough that it does not matter if the resolution is higher than needed in the troposphere.

3) I am unclear on the conversion of atmospheric profile quantities (temperature, pressure, composition) to the (presumably?) equivalent homogeneous paths represented by the kCompressed data. Are these absorber-weighted equivalent (ie Curtis-Godson) temperatures and pressures? Or simply layer means? Given the use of a spherical atmosphere (what radius of curvature is assumed?) rather than a plane-parallel assumption, presumably some sort of numerical integration scheme is required to obtain these quantities, even the total amount of absorber in a layer.
Line 455 of the original manuscript states that our kLAYERS program takes in $n$-level point profiles (typically pressure, temperature, water vapor and ozone profiles), interpolates these points onto a fine grid and then integrates to produce the final integrated layer output profiles for the gases (in molecules/cm2) and one average temperature (same for all gases) at each of the AIRS 100 layers. We wanted to define one layer temperature for all gases, as kCARTA (and SARTA) are used for atmospheric retrievals. Furthermore kCARTA is effectively monochromatic, so does not need Curtis-Godson temperatures.
Internally kCARTA does have the ability to use individual (Curtis-Gordon) gas temperatures at each layer, but those need to be provided as input.
The kLAYERS program uses Planet Earth parameters (radius and height dependent gravity), and also adds in the variable trace gas profiles as needed (such as $CO_2$, $CH_4$, CO). The kCompressed database is saved at the same AIRS 100 layers, so the default kCARTA ingests the output of kLAYERS, un-compresses the

database and does the radiative transfer.

4) Section 4, on the impact of spectroscopy on TOA radiances, seems an unnecessary digression. While spectroscopic uncertainties are certainly an issue that merits attention, that's not really anything to do with the kCARTA model being introduced (besides which all the variation is handled in the generation of the kCompressed datasets, which are to some extent independent). The data gap in the plots, arising from the gap in AIRS coverage, is also undesirably for such a comparison. It would have been more useful to see comparisons of kCARTA TOA radiances against other models instead.
Showing TOA BT uncertainties due to spectroscopy is very important, both for scientists working on retrievals and for those working on data assimilation. To our knowledge this assessment has not been documented (at least recently). However we agree it is an unnecessary digression in this paper, and have removed the entire Section 4 of the original submission.

5) For most molecules, kCARTA uses data created by the UMBC-LBL model using the Van Vleck and Huber lineshape. Since the Voigt lineshape is very much the 'standard', there should be some explanation of how this differs and why it is used in preference to Voigt.
UMBC-LBL is based on GENLN2, which also uses the Van Vleck/Huber lineshape. Initially used with Lorentz lineshapes to model microwave absorption, our implementation is a sum over two voigt lineshapes, one centered at $v_0$ and the other centered at $-v_0$. The VVH lineshape would have larger impacts for the microwave regions; for the infrared wavelengths considered here the second term is negligible and it is essentially the Voigt lineshape.

6) The inclusion of non-LTE effects, just for the CO2 4um band, seems to require the inclusion of a separate line-by-line model within kCARTA. This is a huge overhead in complexity for a relatively specialized application. Given the kCARTA structure, it seems a more natural approach would have been to incorporate vibrational temperature as an extra axis on the kCompressed datasets, and just use GENLN2 to calculate these. Alternatively, if you are including a LBL model within kCARTA, at least extend it so that LBL (with or without non-LTE) can be used for any molecule.
We have already included links to our monochromatic (Matlab based) LBL code. For nadir sounders that kCARTA is designed for, we only need 4 $\mu m$ $CO_2$ NLTE effects where the different vibrational temperatures means we need to account for both the changes in optical depths and for the modifiers to the Planck function. The exisiting code inside kCARTA can compute both NLTE and LTE effects (for the 4 um $CO_2$ lines for which we include line and line-mixing parameters), but since we do not use accelerated Voigt functions or continuums, using it for all molecules would unnecessarily slow down kCARTA. We have explored making a NLTE compressed look-up scheme for kCARTA, for both the optical depths and multipliers to the Planck function. However we did not get satisfactory results, and left this as a "to be revisited someday" project.

7) The calculation of background thermal radiation (section 7) uses an interesting idea, and would probably have merited some expansion as a separate (if somewhat specialized) article by itself. I have a number of questions, detailed in the minor comments
We have answered those questions.

TYPOS/MINOR COMMENTS

P1 L20: '... data are presently ...'
Fixed

P1 L20: Given Susskind was describing cloud clearing for the previous generation of meteorological instruments I'm not sure it's an appropriate reference for the hyperspectral instruments.
The previous generation of weather sounders included HIRS. AIRS is the first of the new generation of hyperspectral sounders. The referenced 1998 Susskind paper is the most relevant reference for operational cloud clearing, and it discusses retrievals using simulated AIRS spectra.

P1 L20: The text may be read as implying that there is some sort of correction applied to the spectra to remove the influence of clouds, whereas I suspect it is more accurate to say that any cloud-contaminated spectra are simply rejected.

With $\sim 15$ km footprints, less than 4% of current hyperspectral sounder observations can be identified as mostly cloud free. The $\geq 60\%$ global coverage retrieval yield of the operational NOAA CrIS NUCAPS and NASA AIRS L2 retrievals is achieved by explicitly using cloud cleared radiances.

P2 L27: What determines this 0.0025cm-1 figure? As a rough estimate I expect it would be determined by the requirement to resolve the Doppler widths of lines, and given molecular velocities are of the order of c/10^6 that would correspond to around 0.001 cm-1 at 1000 cm-1.

As noted at the beginning, and is explicitly part of the new Section 7 in the revised manuscript, the key requirement is that the monochromatic kCARTA radiances can be accurately compared to any real (or hypothetical) sounder radiance. This is achievable using a database generated at 0.0005 cm$^{-1}$, five point box car generated to 0.0025 cm$^{-1}$. However to further improve the accuracy, we have changed the resolution of kCARTA to be 0.0005 cm$^{-1}$ in the 605-880 cm$^{-1}$ cm-1 region. If required kCARTA can easily switch to use higher resolution databases across any spectral range, which now are much faster to generate.

P2 L27: I am not convinced by the argument that the computation of optical depths at high spectral resolution for 50 or so profiles for the training set is something that needs accelerating. Surely this is something that only has to be done once and, even if occasionally repeated, the fact that it takes a week rather than a couple of hours isn't really be an issue.

Our historical motivation has been further explained in Q2 above. kCARTA is also used to check the newly developed (and/or existing) fast model against tens of thousands of other regression profiles we have, as well as tens/hundreds of thousands of AIRS, CrIS and IASI observations as needed. All this can be done extremely rapidly in an embarrassingly parallel fashion with kCARTA. We also generate monochromatic jacobians and weighting functions for some of these observations or test profiles, which for the 605-2830 cm$^{-1}$ region can be done by kCARTA in less than two additional minutes per profile.

P2 L40: I assume the issue with replacing line-by-line with kCompressed data is one of the accuracy of the absorption coefficient or reconstructed layer optical depths. However to state that the 'radiances' are accurately reproduced requires a whole new set of tests to verify the accuracy of the radiative transfer through an atmosphere.

Our 1998 paper stresses we tested the accuracy of the reconstructed radiances against those computed using the uncompressed monochromatic tables. Plus we regularly perform a number of tests offline, involving both single gas and multiple gas radiative path integrals. We hope answers/revisions made to the manuscript further address some of this.

P2 L49: Missing second ')' after 1999.
Fixed

P3 L71: I have not come across the Van Vleck and Huber lineshape. Is this different to the more conventional Voigt? If so, how, and why the unconventional choice?
As explained above, for the infrared it is essentially the Voigt lineshape.

Fig 1: I appreciate this is just a sketch, but the black Total line doesn't seem to be a sum of the red, blue and green lines. Also raise the '-1' in the x-axis title to a superscript (and in subsequent figures).
We have fixed the xlabel here and in other places. Regarding the sums, we checked that everything is OK by modifying the code that generated this plot to print out the $y-$ values of the blue, green and three red curves at various points inside the $x \in (-0.5, +0.5)$ interval, the sum of these values, and the $y-$ value of the black curve, and verified they were identical.

P4 L98: Regarding MonoRTM - what is the point being made here?
Our understanding is that monoRTM is the reference line-by-line code which LBLRTM is checked against. So, our compressed ODs are as accurate as the monoRTM ODs (at least for 10 um $O_3$ absorption spectrum), and we are confident the Van Vleck/Huber lineshape in the UMBC-LBL code will work just as accurately when used for appropriate molecules in the IR region (this would obviously not be true for molecules that use specialized lineshapes, such as $CO_2$ and $CH_4$ linemixing). Since that point did not come across clearly, and is not really needed, we have removed it.

P5 L107: +/- 50K does not seem a large range in temperature. Do you have any evidence that it spans

We read in one set of ECMWF data for 2019/08/01 (360x180 one degree grid points) and ran the $\sim 64000$ profiles through kLAYERS. All but 4% of the temperature profiles lay within $\pm 50$ K of the US Standard temperature profile. The ones that lay outside these bounds were all profiles over the Antartic plateau, on average $3 \pm 2$ K outside the -50 K offset (between 500 - 1000 mb). kCARTA handles these cases by extrapolating compressed ODs (and zero checking) as needed. We have added this information into the appropriate place of the revised manuscript using the following phrases "Tests using NWP profiles show this is usually sufficient everywhere except for a handful over the winter Antartica, which could fall slightly outside the coldest offset (on average by about 3 K) between 600-1000 mb; kCARTA handles these cases by extrapolating what has been compressed."

P5 L108: 'contains'
Fixed

P5 L122: HITRAN 2016 lists two further isotopologues for water vapour containing a single deuterium atom and either a 17O or an 18O oxygen isotope (these are HITRAN isotopologues 5 and 6). Are these included with HDO or with the remaining H2O isotopologues?
With the remaining H2O isotopologues

P6 L144: It is not clear why the cross-section molecules are also represented using kCompressed databases. Presumably these end up much larger than the original files, which usually have only a few tens of (pressure,temperature) points and a much coarser spectral axis. Also using CIA probably won't work with LUTs - how will these new data be used?
We chose to do it this way in order to compute the ODs of all gases (molecular and cross-section) equally. In any case our database size is dominated by the main molecular gases ($H_2O$, $CO_2$, $O_3$). The CIA is handled by calling the necessary routines within kCARTA.

P6 L150: HITRAN 2016 lists 49 rather than 42 molecules, and a number of these (or even 1:42) are not represented in the US Standard Atmosphere.
Correct, we only use the first 42 as we were able to get the "standard" or "realistic" profiles for them; similarly now HITRAN has very many cross sectional profiles but we only use the ones for which we are able to find representative profiles in the scientific literature.
We have amended the sentence to read "The default kCARTA mode is to use the 42 molecular gases in HITRAN database, together with about 30 cross-section gases, for which we have reference profiles. "

P6 L158: 'Schwarzschild'
Corrected

P7 Eq(3): The solid angle integration should just be over a hemisphere and should include the B.dt/ds term scaled by cos(e) where e is the elevation angle 0:pi/2 in the hemispheric integration (thus the integral of cos(e).dOmega from 0:2pi on its own should yield pi).
Both fixed, thanks for pointing out these mistakes

P7 Eq(3): I don't see why the cos(theta_sun) appears in the last term on its own, but it seems there should be some solid angle integration over the sun's disk (as in Eq 6) otherwise it will be as if the whole sky radiates at the solar temperature.
The manuscript has defined $B_{\odot}(\nu)$ as the solar radiance at the TOA, so that accounts for the solar disk.

P7 L170: Isn't 1-epsilon_s(nu) the same as rho_s(nu)?
Default behavior of kCARTA is to do this; however we can explicitly input reflectivity so that we could for example handle sun glint off an ocean

P7 L171: extra comma near end of line
Fixed

P7 L171: There is no tau_i term in Eq(3), just tau and tau_atm.
Fixed

P7 L180: Assuming the temperature profile is specifed at points P,H1,H2 etc what constant temperature is assumed for, eg the lowest layer? Is it T(P), T(H1), or something else?

Already answered above when we respond to the question regarding use of kLAYERS; it is the layer averaged temperature for the layer between $P$ and $H1$

P7 L182: By 'density effects' do you mean refraction?

Correct

P7 L186: Both emissivity *and* reflectance have to be supplied? Eq(4) only uses emissivity.

Yes, Equations (3,6,7) shows that kCARTA also uses reflectivity. We have amended both sentences in that and subsequent sections to make it more clear.

P8 Eq(5): The indexing doesn't seem to work. Interpreting tau_(i+1 to N) as the transmittance from the base of layer i+1 to the base of layer N the calculation for layer 3 in the diagram would be ( 1 - tau_3 ).tau_(4 to N) but here N=4 so tau_(4 to N) = 1 whereas it should be the transmittance through layer 4. Similarly Eqs 8-10

We see the confusion has arisen because we forgot to state that (a) $\tau_i$ represents the transmission through layer $i$ (i.e. from bottom to top of layer $i$), and that (b) $\tau_{i+1 \rightarrow N}$ is the transmission from bottom of layer i+1 to top of layer N. So we have taken the opportunity to add in these definitions and make some additional clarifications in the relevant text/equations.

P8 L195 Better to swap sections 3.3 and 3.4 to match the same order these terms appear in Eq (3)

Done

P8 L196 rho is now defined as reflectance, but for Eq 3 it was reflectivity. Is there a difference?

We now consistently use reflectivity everywhere, instead of reflectance.

P9 L213 As a general comment, it would be nice to have a plot of the magnitude of these four terms as a function of the infrared spectrum, assuming say some fixed surface emissivity of around 0.98 (so 2% diffuse reflectance).

We added in a number of figures detailing the comparisons against LBLRTM, and decided not to do this.

P9 Section 4: Presumably for this exercise different sets of kCompressed databases were computed by running UMBC-LBL etc for the different sets of spectroscopic data, and then running kCARTA using these 3 different sets of kCompressed databases. If one really wants to demonstrate the differences in TOA radiance couldn't one simply run the LBL models with the different spectroscopic data and eliminate the whole intermediate step of generating kCompressed datasets?

We re-iterate that running the UMBC line-by-line code which does not use acceleration for the Voight fucntion and/or gas continuums is very time consuming, as it partitions the lines into "near" "medium" and "far". This has to be done molecule by molecule, layer by layer, across the entire 605-2830 cm$^{-1}$ spectrum. This is a significant amount of time even if computed in embarrassingly parallel mode. Our 1998 JQSRT paper already shows how accurate our compressed database is. This means once the compression is done, kCARTA can be used to generate synthetic radiances for thousands of NWP model atmospheres in a number of minutes (when kCARTA is used in embarrassingly parallel mode).

P9 L230 'linemixing'

Fixed

P10 L243 'sagain' ?

Replaced with "panel"

P10 L247 it is not clear that differences have anything at all to do with linemixing - it seems they might simply reflect differences in the standard line widths that would be evident whether line-mixing effects were included or not?

That is possible, but we cannot exclude that differences in mixing coefficients determine how much intensity has to be transferred from the wings to the peaks. We have not investigated that idea as for now we have chosen to simply use available $CO2$ codes. A proper test would involve using other line-mixing codes together with different spectroscopic databases, but that is outside the scope of this paper.

P13 Section 5: It seems odd that kCARTA has a non-LTE line-by-line module - it is something I would have expected in the UMBC-LBL code. Have you considered modelling non-LTE using kCompressed datasets? That would seem more in keeping with the overall design. Perhaps you would need an extra tabulated dimension in vibrational temperature, or (vib-kin) temperature?

Our line-by-line code could indeed be modified to generate the ODs using the vibrational temperatures. However NLTE also effects the Planck function and we would also need to compute the multipliers to the Planck function and give them to kCARTA. For these and other reasons it is more natural to put the NLTE effects directly into the kCARTA RTA. As explained above, generating compressed lookup tables for NLTE effects remains a "to be revisited someday" project.

P13 Eq 9: Summations should be from i=1,N. Also, tau_(i+1 to N) in second summation should have (nu) afterwards.

Fixed

P14 L304: 'Jacobian' from here to the end of the section start to be capitalised - inconsistent with earlier 'jacobian'

Fixed everywhere

PP14 L305: How is dB/dT calculated? (where s_m = T_m).

Analytic derivative of the Planck function

P14 L310: I don't understand the last sentence - what's the difference between the Jacobian and the weighting function wrt surface temperature and emissivity?

We have rewritten the sentence to state "kCARTA also computes the weighting functions, and jacobians with respect to the surface temperature and surface emissivity."

P14 L320: 'contributes'

Fixed

P15 L334: the dmu should come after the exp(-x/mu).

We have moved $\mu d\mu$ after the $exp(-x/\mu)$

P14 L334: Is there any significance in labelling this integral as E_3 ?

It is the exponential integral of the third kind, and have added this to the text.

P14 L334: No closing bracket to match '('.

Fixed

P14 Eq 13: From discussing downward radiation, the introduction of reflectance rho_s suggests to me that you are now modelling the reflected component of this radiance just above the surface. Where does the 2pi come from? I think you need to explain some of the intermediate steps.

Assuming azimuthal symmetry when doing the hemispheric integral gives the factor of $2\pi$.

P14 L340: If I understand this correctly, instead of approximating the downwelling radiance as if it comes from a fixed angle cos=3/5, you are adjusting the angle for each atmospheric layer according to the surface-layer optical thickness x. It would be nice to see some plots of the comparisons with a full hemispheric integration to show that this is sigifnicantly more accurate than the fixed angle assumption. Also, is there a reason why the cut off at 30deg has to be applied for optically thick atmospheres? Otherwise it might also be useful for modelling the radiance viewed by upward-looking instruments, or downward looking instruments close to the surface.

LBLRTM does flux calculations at 3-4 gaussian quadrature angles, which is evidence that a single angle asumption is not accurate enough. Instead of this, we chose to do downwelling background thermal using a varying diffusivity angle at each layer. For optically thick regions, a TOA sounder is not going to remotely sense any contribution from the surface, whether it is directly emitted by the surface or reflected from the surface. Hence, in these regions there is no need to calculated the background thermal at all. We agree that for the more transperant regions, a little more care should be taken for downward looking instruments close to the surface, including a finer layering of the atmosphere closer to the surface, which kCARTA and kLAYERS can both handle, as discussed in the 1998 paper.

 Given the rapid spectral variation in any 25cm-1 interval, I don't understand how you can assign a single assumption to the whole interval. Won't there be a whole range of optical thickness within the 25cm-1 region so that the assumption works better for some spectral points than others?

Both the lines and wings of an optically thick region are mostly opaque i.e. you do not go from transparent to optically thick in a few tenths of a wavenumber, but rather over an appreciable interval. So encompassing $25 \text{ cm}^{-1}$ chunks as we did is fine. Furthermore we have tested our assumption against Gaussian quadrature, and our method is far superior to simply using acos(3/5) for downwelling radiation. The user can also opt to use only acos(3/5) or do Gaussian quadrature.

P15 Sec 7.2: The linear-in-tau model, where optical depth is scaled by the sec(theta) to allow for off-nadir viewing angles, assumes a plane-parallel atmosphere where theta is fixed for the layer (and the same for every layer). How is this handled for the spherical atmospheres assumed in kCARTA where cos(theta) can vary significantly when viewing off-nadir?

kCARTA defaults to dividing an 80 km thick atmosphere into about 100 layers, with the layers starting out being about 0.25 km thick at the bottom, and gradually increasing in thickness the higher you go. The linear-in-tau models the temperature variation through each of the individual layers. So kCARTA does linear-in-tau T(tau(i)) at angle theta(i) the same way as it does constant T(i) at angle theta(i) : by varying the angle layer by layer as the beam propagates upwards.

P16 L365: It depends what you mean by the 'average layer temperature'. For the optically thin limit you would expect this to converge to the Curtis-Godson temperature, ie the absorber-weighted mean temperature, which would generally be at an altitude below the layer mid-point.

As explained above, we accurately determine the mean layer teperature using kLAYERS, and then use the definitions in the Clough et al 1992 JGR paper to determine the temperature variation across the layer.

P16 L377: Even though it largely disappears after convolution (presumably because the AIRS spectral resolution means that the radiance is dominated by contributions from lower altitudes), this 10 K difference does seem to be a serious issue. And one which would limit the use of kCARTA for accurately representing finer resolution instruments. Is this really due to not implementing linear in tau or could it be that kCARTA only uses a crude representation of layer temperature rather the something more physically justified such as Curtis Godson temperature? Why doesn't kCARTA just use the linear-in-tau model?

Given an input temperature profile, kLAYERS internaly interpolates to a fine grid before accurately finding the mean layer temperature. We do not need a Curtis-Gordon temperature since at each wavenumber point we are essentially monochromatic (not a band model). It is also well known that hyperspectral sounders have at most about 12-15 degrees of freedom for temperature, so our 100 layers are more than adequate. We also input the kLAYERS temperature profile into the LBLRTM TAPE5, and as far as we can tell it is then not using a more physically justified temperature than kCARTA does.

As mentioned in the title and at the beginning of the answers, kCARTA is a pseudo (monochromatic) line-by-line code. The large (10+ K) differences are seen when comparing kCARTA to LBLRTM at 0.0025 $\text{cm}^{-1}$ resolution, since the latter internally is doing the upper atmosphere calculations at high resolution. As we improve the kCARTA database resolution, the differences become significantly smaller. For example if we use 0.0002 $\text{cm}^{-1}$ resolution, it drops to less than 1 K $\pm$ 1 K right on top of the high altitude 15 $\mu m$ $CO_2$ lines, and -0.1 $\pm$ 0.05 K in the high altitude $O_3$ sounding channels, when averaged over our 49 regression profiles. If we use 0.0005 $\text{cm}^{-1}$ resolution, the 15 $\mu m$/10 $\mu m$ numbers are correspondingly 4 $\pm$ 1 K and -0.3 $\pm$ 0.1 K. After convolution with a sounder SRF, the differences are neglible. As mentioned earlier, we thank the referee for pointing this out to us and have made 0.0005 $\text{cm}^{-1}$ the default resolution in the thermal IR; any interested user can easily generate and use a higher resolution grid if desired.

We note that it appears that we run into slight differences in $CO_2$ line broadening and/or resolution right on top of the lines, and perhaps algorithm differences (LBLRTM may use a Pade approximation and/or Eqn 15/16 to first order while we use Eqn 16 to fifth order).

The above evidence provides ample confidence that the linear-in-tau RTA is working quite well, even when allowing for ray tracing. As expected after convolution with a sounder SRFs, these differences mostly vanish, since these differences are right at the peaks of a small number of very high sounding $CO_2$ lines.

This is all described in the (new) section 7, on inter-comparing kCARTA and LBLRTM. We note this meant

we also had to change a few sentences in the section on flux computations, and in Appendix B. kCARTA now uses the linear-in-tau model.

P17 L390 'computes'
Fixed

P17 Flux Computations: are these with spherical or plane-parallel atmospheres?
Whether you use the (exponential or legendre) gaussian quadrature, we use the same fixed quadrature points at each layer so that is plane parallel.

P17 L407: Given the differences found from the linear-in-tau model in the previous section, could the differences in heating rates at high altitude simply be another manifestation of the same problem, ie assumption of constant temperature within thick layers at higher altitudes?
We believe our above responses above adequately address this issue, namely it is the resolution. In addition we have stated when doing flux computations, kCARTA uses linear-in-tau.

P18 Fig 6 caption: should be '0.0005' instead of '0.005'.
Fixed

P18 L425: 'accuracy of its spectroscopic database'. I assume this refers to the kCompressed tables used as kCARTA input as opposed to the usual meaning which would be HITRAN or GEISA spectroscopic databases. But comparisons against GENLN2 or LBLRTM wouldn't just be a comparison of the kCompression with the original HITRAN unless your tests were for simple homogeneous paths where transmittance could be verified independently of other model assumptions such as ray-tracing and integration through atmospheric layers.
Correct we are referring to the accuracy of the compression.

P19 L434: '0.0025 cm-1 is good enough for nadir hyperspectral sounders' - this is a contentious statement, and needs some justification (there is none in this paper).
We are confident our responses to the earlier questions address this issue, especially in light of the fact that most residuals are far smaller than detector NeDT when the radiances are convolved with realistic sounders response functions.

P21 Table B2: lists (6) Direction as downwelling by default, upwelling as an option. Isn't it the other way around?
Corrected, thanks for pointing this out

P22 L488: extra ')'.
Fixed

P22 L495: 'up to'
Fixed

Reviewer 2

*Specific comments*

1). What studies have the authors done to ensure that the 11 temperature grid points are adequate to represent the temperature dependency of optical depths of gases for each atmospheric layer. Please quantify the errors of interpolation due to 11 point grid and the choice of interpolation method (spline vs linear).
We have addressed the first part of this question above. The spline versus linear interpolation errors work out to be $0.0004 \pm 0.0040$ K when averaged over all monochromatic spectral points, for 49 regression profiles, with a maximum absolute difference of 0.342 K (in the 15 um region). This information has been inserted into Appendix B of the revised manuscript.

2). The kCARTA package is optimized for the thermal infrared spectral region. Though the authors claim it is trivial to extend the database out to span the far infrared to ultra-violet range, the package does not include Rayleigh scattering and an accurate multiple scattering radiative solver, which are important for the shortwave top of atmosphere radiation calculations.
Correct, we have fixed Section 9 and Appendix to state that PCLSAM is optimized for the thermal infrared away from solar scattering effects, so we have codes to read in ODs uncompressed by kCARTA for arbitrary atmospheres, which can then be passed to for example LBLDIS.

3). On page 2 line 40, it will be useful to describe the relative errors between kCarta and the MNLBL.
The UMBLBL code computes only optical depths, and does not have any radiative transfer computations. As mentioned above and in the next question, we have added a section that describes detailed top-of-atmosphere radiance intercomparisons against LBLRTM.

4). Section 2.1 on page 4. Although the authors described the UMBC-LBL model line shape calculations and mentioned that extensive comparisons with LBLRTM and GENLIN2 have been performed, no quantitative results have been shown to illustrate differences among different LBL models. For some molecules, a sub-Lorentz line shape is used by LBLRTM, what about the UMBC-LBL?
As asked by two reviewers, we have added a section that describes detailed inter-comparisons against LBLRTM. We are also happy to provide interested users with a database generated entirely using the optical depths in LBLRTM.

5). The method used by kCARTA for calculating the downwelling background radiative is very efficient and much more accurate relative to a constant diffusive angle. However, for a non-Lambertian surface (e.g a specular reflection surface), this may not be a good approximation.
Correct, an example BRDF for sunglint is modeled in Appendix A of the Nalli et. al 2016 paper we have referenced.

There are some minor errors in the manuscript:

1. In the figure caption for Figure 6, 0.005 cm-1 should be 0.0005 cm-1.
Fixed

2. Not all the symbols used in this paper are defined. For example, Omega in equation (3), g under equation (C1). . .
Fixed

3. Some of the links given in Appendix A are not available.
Fixed the line-by-line code link to point to https://github.com/ sergio66/UMBC_LBL. The others were valid/accessible.

4. Line 149, the symbols in the equation are not explained by the text or the appendix B.
Fixed

5. Line 230, please correct typo "limemixing"
Fixed

*Specific comments*

1) This paper describes a particular incarnation of kCARTA which the authors are making available. It would be very helpful to have a version identifier for the version described in the paper, and perhaps further identification, if possible, of which previous versions have been widely used. This is particularly needed in the paragraph beginning at Line 132, where the phrase "we now use" is unclear as to which previous version is obsolete.

In Appendix A we now explicitly mention that SRCv1.21, together with HITRAN 2016 and LBLRTM12.8 for $CO_2$, $CH_4$ is used for the latest coefficents for the SARTA v2.01 (2019) versions for AIRS, CriS and IASI. In the same location we also clarify that we generate a new compressed database every 4 years (roughly within a year of a new HITRAN release)

Typographical errors:

Line 28, add comma after "process"
Fixed

Line 59-60, mismatched parentheses
Fixed

Line 158, "Schwarzschild" is misspelled
Fixed

Line 163, "radiation propagating" are both misspelled
Fixed

Line 171, remove the extra comma
Fixed

Line 242, unmatched bracket
Fixed

Line 243, "again" misspelled
Fixed

Line 322, remove comma after "i"
Fixed

Line 334, mismatched parentheses
Fixed

Line 335, coming "from" all
Fixed

Line 493, "cloudy" should be "cloud"
Fixed

Line 536, "A.B." should be Boynard, A.
Fixed

Line 536, "P.M." should be Pommier, M.
Fixed

Line 536, "A.R." should be Razavi, A.
Fixed

Line 537, "Chenm" should be "Chem"
Fixed

Line 546, "Iancono" should be "Iacono"

Fixed

Line 580, "humidity" misspelled
Fixed

Line 584, list of authors is incomplete, at least add et al.
Fixed

Line 613, "I.E. G." should be "Gordon, I.E."
Fixed

Line 632, "Wm." should be removed
Fixed

Line 649, "Karlsruhe", not "Karlsruhr"
Fixed

Line 649, this link is dead. I find the paper at https://www.imkasf.kit.edu/downloads/SAT/kopra_docu_part02.pdf
Fixed

List of changes to manuscript

Here we list the changes to be found in the revised manuscript, and leave out fixes to typographical/spelling errors found in the original manuscript.

*Major changes* : The first two are the major changes, and are mentioned prominently in the abstract, introduction, Section 7 and conclusions.

- The default resolution of kCARTA is now 0.0005 cm$^{-1}$ in the 605-880 cm$^{-1}$ region, and 0.0025 cm$^{-1}$ in the 805-2830 cm$^{-1}$ region

- The default temperature variation is now "linear in tau"

- new Section 7 describes top-of-atmosphere Brightness Temperature intercomparisons between kCARTA and LBLRTM

- We have removed Section 4 from the original manuscript, which described the impact of spectroscopic uncertainties on synthetic top-of-atmosphere radiances

*Changes/fixes in response to reviewer comments*

- Section 2.2 (lines 115-118) : we describe the applicability of the $\pm$ 50 K temperature offsets to realistic NWP model temperature profiles.

- Section 2.2 (lines 119-121) explicitly state the default resolution of kCARTA

- Section 2.2, (end, lines 160-165), we have added definitions of $\nu_0, T, m$ and clarified which gases are used in kCARTA

- Section 3, page 7-8 : we have fixed the equations and clarified some of the sentences describing the clear sky RTA in kCARTA

- Page 9, Sub-sections on background thermal and solar radiation have been swapped

- Equations in Section 5 (Jacobians) have been fixed, and some sentences clarified.

- Manuscript consistently uses reflectivity and emissivity everywhere

- Section 6.1, line 296 : Background thermal section defines $E_3(x)$

- Section 6.2 : Made some changes since kCARTA now uses default "linear in tau"

- Section 7 is a new section, describing in detail top-of-atmosphere brightness temperature differences between kCARTA and LBLRTM, as averaged over 49 regression profiles. The section describes these BTDs both monochromatically (as a function of varying database resolution), and after convolution with a typical hyperspectral sounder response function.

- Section 8 (flux comparisons) has been slightly changed since now we use as default a higher resolution database.

- Section 10 (Conclusions) has been slightly changed since we now use a higher resolution deep in the 15 $\mu m$ region, and "linear in tau" temperature variation.

- Appendix A (code and database availability) has also been changed to reflect the above, and includes a short discussion of the version used to generate the current (2019) SARTA fast model database.

- Appendix B (kCARTA features) has been edited to reflect the above changes

- Appendix C (scattering) we now emphasize kCARTA does not have built in SW scattering ability, and suggest routines we have to output tables that can be read by for example LBLDIS. We have also made some changes to the text as requested by the reviewers.

- Acknowledgement section has been edited

- References should be free from mistakes.

[revised manuscript text omitted]

---

## Author Response (AR2)

Sergio DeSouza-Machado,
Department of Physics,
University of Maryland Baltimore County,
1000 Hilltop Circle,
Baltimore MD 21250

November 27, 2019

Dr. Lars Hoffmann,
Associate Editor,
Atmospheric Measurement Techniques

Dear Dr. Hoffmann,

This cover letter is included with the submission of final manuscript for publication (amt-2019-282) entitled

"kCARTA : A fast pseudo line-by-line radiative transfer algorithm with analytic Jacobians, fluxes, Non-Local Thermodynamic Equilibrium and scattering for the infrared"

In writing the manuscript, as before we have adhered as strictly as possible to the AMT manuscript guidelines. This document includes

- Page 1 : this cover letter,
- Pages 2-27 : output from latexdiff between the November 19, 2019 manuscript and today's submission

I am also separately uploading the required files to AMT. Should you need to contact me, my email address is sergio@umbc.edu.

Sincerely,

Sergio De Souza-Machado

[revised manuscript text omitted]